# TOWARDS DYNAMIC INTERLEAVING OPTIMIZERS

**Yile Chen**[1,2], **Zeyi Wen**[2,3,*], **Jian Chen**[1,*], **Jin Huang**[4]
[1]South China University of Technology, [2]HKUST (Guangzhou), [3]HKUST,
[4]South China Normal University
`jireh.x6@gmail.com, wenzeyi@hkust-gz.edu.cn, ellachen@scut.edu.cn,`
`huangjin@m.scnu.edu.cn`

## ABSTRACT

Optimizers are critical for training deep neural networks. Existing training processes rely on a single static optimizer (e.g., SGD) or a simple hybrid of two optimizers, which miss the opportunity to exploit evolving dynamics in different training states, degrading model quality and convergence. In this paper, we propose a novel dynamic optimizer switching method called *Dynamic Optimizer Interleaving Training* (DOIT) method, which builds surrogate models to predict different optimizers' performance from current parameter states. DOIT uses an acquisition function that combines the results from surrogate models with transferability assessments and process information to select a suitable optimizer for the subsequent training. Experiments on various models and tasks (e.g., image and text classification, machine translation, and object detection) show that DOIT effectively enhances the training, achieving faster convergence (i.e., 2% to 10% faster) and higher accuracy (i.e., 1% to 3% improvement). Additional independent experiments and case studies further validate DOIT's effectiveness.

## 1 INTRODUCTION

The choice of optimizer and its hyperparameter settings can significantly influence both model quality and convergence in training deep neural networks (DNNs) (Soydaner, 2020; Hassan et al., 2023). Researchers have proposed various optimizers for better model training (e.g., SGD and AdamW), and have found that different optimizers exhibit distinct characteristics and are suited to different tasks. For instance, SGD tends to offer better generalization and is often used for head fine-tuning (Poojary & Pai, 2019), while Adam converges faster and is commonly used in LoRA (Hu et al., 2021). Moreover, considering the distinct strengths of various optimizers, some researchers have explored combining them to enhance training performance (Keskar & Socher, 2017).

However, recent studies have found that different optimizers not only behave differently on different tasks, but also exhibit unique characteristics and optimization strategies in different training states (Im et al., 2016). For example, Figure 1 presents the optimization results of three optimizers with varying runs and hyperparameter settings on the Three-Hump Camel function, revealing that: "*different optimizers are suitable for different training states (or parameter states)*". Static optimizers and simple hybrid methods struggle to exploit the training dynamic, often leading to unstable model quality and increased training costs (Sun, 2020).

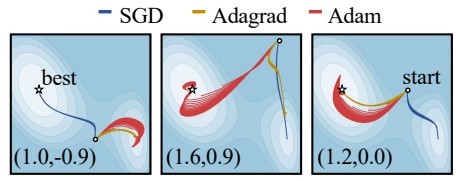

Figure 1: The different training processes with various optimizers.

To address this problem, we propose **D**ynamic **O**ptimizer **I**nterleaving **T**raining (DOIT) method, which dynamically selects the suitable optimizer according to the current training state. Throughout the training process, DOIT constructs and continually updates surrogate models for different optimizers to estimate their optimization performance under various parameter states. Meanwhile, DOIT calculates a score for each optimizer using a special acquisition function that combines the

---

*Corresponding Authors

estimated performance, a transferability assessment, and training process information. By carefully selecting the optimizer with the highest score during training, DOIT achieves faster convergence and better model quality. To summarize, the key contributions of this paper are as follows.

- We investigate the distinct strengths and optimization directions of various optimizers across different parameter states. Furthermore, we demonstrate that combining different optimizers during training can help achieve higher-quality models and better convergence.

- We propose Dynamic Optimizer Interleaving Training (DOIT), a method that adaptively switches optimizers based on parameter states during training. DOIT estimates optimizer performance under different parameter states by constructing Gaussian surrogate models and calculates scores using a well-designed acquisition function. By iteratively selecting the optimizer with the highest score, DOIT produces higher-quality models with faster convergence.

- We implement DOIT and conduct experiments on various models and tasks, including full training and PEFT. Experimental results demonstrate that DOIT effectively improves the training performance, achieving faster convergence (i.e., 2% to 10% faster) and higher accuracy (i.e., 1% to 3% improvement). Furthermore, we present additional case studies and independent experiments to further validate and analyze the performance of DOIT.

## 2 RELATED WORKS AND BACKGROUND

In this section, we provide the background of our work, including optimizers, simple hybrid methods, and AutoML. After that, we identify the limitations of existing approaches.

**Optimizers.** The optimizers and their hyperparameters are crucial for training DNNs. Traditional gradient descent computes gradients over the full dataset and updates parameters accordingly (Ruder, 2016). Building on this, various optimizers have been proposed, such as momentum (Sutskever et al., 2013) and Nesterov Accelerated Gradient (Qu & Li, 2019). Figure 2 illustrates a portion of the historical development of these optimizers. Beyond these, researchers have also proposed various second-order optimizers, such as L-BFGS (Liu & Nocedal, 1989), K-FAC (Martens & Grosse, 2015), and AdaHessian (Yao et al., 2021). However, due to their practical application challenges, the second-order optimizers are not further discussed in this paper. Additionally, recent work has further introduced learned optimizers via meta-learning (Harrison et al., 2022).

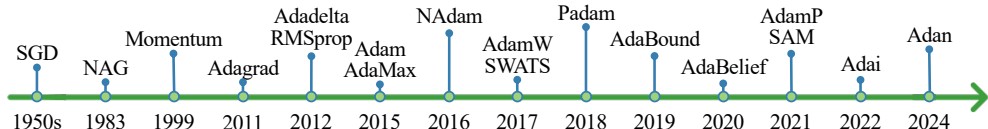

Figure 2: The development of neural network optimizers.

**Hybrid optimizers.** In addition to in-depth studies on individual optimizers, researchers have also proposed simple hybrid methods that integrate different optimizers-typically combining the strong generalization capability of SGD (Sutskever et al., 2013) with the rapid convergence offered by adaptive learning rate methods (e.g., Adam (Kingma & Ba, 2014) and Adagrad (Duchi et al., 2011)). For example, SWATS (Keskar & Socher, 2017) switches from Adam to SGD for better results. Padam (Chen et al., 2018) blends the two using a partial adaptive parameter, while AdaBound (Luo et al., 2019) uses dynamic learning rate bounds for smooth transitions. In addition, AGD (Yue et al., 2023) proposes a switch method between SGD and Adam by the preconditioning matrix. In addition, several methods combine traditional optimizers with learned optimizers to address the inherent stability and out-of-distribution challenges of purely learned optimization, such as HUB (Dai et al., 2024) and LGL2O (Prémont-Schwarz et al., 2022).

**AutoML and HPO.** Beyond optimizers, our work is also closely related to hyperparameter optimization (HPO) (Feurer & Hutter, 2019) and AutoML (He et al., 2021). Traditional HPO methods use static configurations, where a fixed set of hyperparameters is used throughout training, such as random search (Bergstra & Bengio, 2012) and Bayesian optimization (Shahriari et al., 2015). More recent studies have proposed dynamic algorithm configuration (Adriaensen et al., 2022), exploring

areas such as genetic algorithms (Biedenkapp et al., 2022) and learning rate control (Daniel et al., 2016). Moreover, our work is conceptually related to meta-learning (Vettoruzzo et al., 2024), but meta-learning focuses on cross-task adaptation, whereas we focus on tuning within a single training.

**Limitations of current approaches.** (i) Static optimizer: While improved optimizers aim to better adapt to parameter landscapes, they often incur high computational costs and may not outperform SGD (Keskar & Socher, 2017). To obtain better models, researchers need to train with different optimizers, which is time-consuming. Additionally, consistent optimizer throughout the entire training process limits both model quality and convergence speed. (ii) Simple hybrid optimizer: Combining different optimizers can enhance model quality and convergence. However, existing methods merely perform a simple switch from one optimizer to another, neglecting the unique strengths of different optimizers under different training states. Such a coarse approach not only restricts the stability of the model quality but also impacts convergence speed (Zhuang et al., 2020).

# 3 OUR PROPOSED METHOD: DOIT

To better utilize multiple optimizers, we propose **D**ynamic **O**ptimizer **I**nterleaving **T**raining (DOIT), a method that enables dynamic optimizer switching during training. We first provide a brief overview of DOIT, followed by a detailed introduction for the key components.

## 3.1 OVERVIEW OF OUR PROPOSED DOIT

Figure 3 illustrates the workflow of DOIT, and a detailed description with pseudocode is presented in Appendix A. DOIT calculates the transferability weight $\omega_t$ to assist in the subsequent selection of optimizers before the training (Step 1). In each optimizer switch cycle of $\tau$ iterations (Steps 2-6), DOIT first compresses the model parameters $\boldsymbol{\theta}^i$ to get the input of the surrogate model (Step 2). Then, DOIT selects the appropriated optimizer $o^i$ for the current training state (Steps 3-4). Obtaining the training losses, DOIT calculates the performance score $s$ and updates the corresponding surrogate model $g^i$ (Steps 5-6). By iteratively executing this process, DOIT achieves the dynamic optimizer switching.

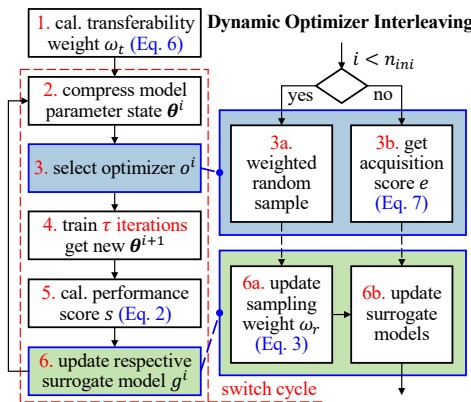

Figure 3: The workflow of DOIT.

For selecting the suitable optimizer (Step 3), DOIT employs two methods: weighted random selection and recommendation based on the surrogate model. In the initial training stages, DOIT uses the calculated score $s$ to update the sampling weight $\omega_r$ for randomly selecting optimizers (Steps 3a and 6a). After acquiring enough training results, DOIT selects the optimizers with the highest score $e$ calculated by the acquisition function for each training stage (Step 3b).

## 3.2 MOTIVATION AND PROBLEM FORMULATION

Before introducing the details of the surrogate model and acquisition function in DOIT, we first provide the hypothesis underlying our method: *"different optimizers offer distinct optimization directions and are suited to different parameter states"*. Figure 4 illustrates examples of the different optimization directions with different optimization functions. It can be observed that although the optimizers start with similar directions in the first iteration, their paths diverge after a few iterations. Previous studies have observed this phenomenon, noting that optimizers exhibit different optimization directions under varying parameter states (Im et al., 2016). Thus, we believe that the optimizer type also requires fine-grained tuning (Adriaensen et al., 2022).

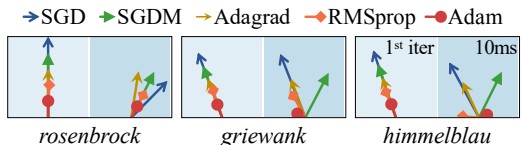

Figure 4: Optimizers offer distinct directions.

Building on this assumption, DOIT attempts to propose a dynamic optimizer switch method that leverages the strengths of different optimizers for distinct parameter states. Let $o \in \mathcal{O}$, $\boldsymbol{\lambda} \in \Lambda$, and $t \in \mathcal{T}$ denote the optimizer type (e.g., SGD), hyperparameter setting (e.g., learning rate as 0.1) and the training time (e.g., 5 iterations), respectively. Then, the training process with dynamic optimizer switches can be defined by a list of configurations $\mathcal{C} = \{c^1, c^2, ..., c^n\}$ where $c^i = (o^i, \boldsymbol{\lambda}^i, t^i)$. The objective of DOIT is to find an optimal $\mathcal{C}^*$ that minimizes the following objective function:

$$\mathcal{C}^* = \arg\min_{\mathcal{C} \in \mathcal{O} \times \Lambda \times \mathcal{T}} \mathcal{L}(\boldsymbol{\theta}^0, \mathcal{M}, \mathcal{D}, \mathcal{C}) \tag{1}$$

where $\boldsymbol{\theta}^0$ is the initial model parameter state, $\mathcal{L}(\cdot)$ is the loss of the trained model $\mathcal{M}$ under dataset $\mathcal{D}$. Equation (1) represents dynamic optimizer tuning during training. When all $c^i \in \mathcal{C}$ share the same settings, it aligns with the traditional training, which is described further in Section 4.1. For clarity, we fix all training times $t \in \mathcal{T}$ to a constant $\tau$ (i.e., 5 iterations) in the following sections.

### 3.3 ESTIMATING PERFORMANCE WITH SURROGATE MODELS

DOIT employs a Sequential Model-Based Optimization (SMBO) to address dynamic optimizer tuning, as shown in Figure 5. Initially, DOIT trains model $\mathcal{M}$ with random configurations to gather experience. It then constructs surrogate models $\mathcal{G} = \{g_1, g_2, ..., g_m\}$ for each optimizer type $o_i \in \mathcal{O} = \{o_1, o_2, ..., o_m\}$ to guide configuration selection. By iteratively selecting configurations and updating surrogate models, DOIT achieves a dynamic optimizer switch training. In this section, we introduce DOIT's surrogate model through its selection, input, output, and initialization.

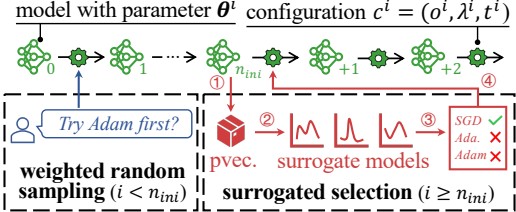

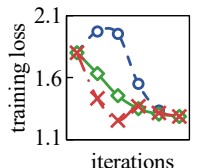

Figure 5: The training process of DOIT.

Figure 6: Examples of the optimization gain.

|  | $o_1$ -O- | $o_2$ -◇- | $o_3$ -X- |
|---|---|---|---|
| $\mu_\Delta$ | 6.1 | 6.4 | 6.1 |
| $\sigma_\Delta$ | 11.0 | 4.0 | 11.0 |
| $\Delta_{upper}$ | $-1.7$ | 1.9 | 4.1 |
| $\Delta_{lower}$ | 5.7 | 5.7 | 6.1 |
| $s$ | 4.5 | 5.1 | 6.5 |

**The selection of the surrogate model.** DOIT adopts Gaussian process (GP) models (Schulz et al., 2018) as surrogate models for several reasons. First, compared to other models, GPs can efficiently train and continuously update the surrogate model as training progresses. Second, GPs provide uncertainty estimation for predictions (i.e., the variance), which is useful for guiding the optimizer selection (as detailed in Section 3.4). Third, as a powerful probabilistic model, GPs effectively construct the overall distribution based on known points, offering flexibility and interpretability.

**The input of the surrogate model.** At the beginning of each switch cycle, DOIT acquires the input for the surrogate model $\text{VEC}^i$. The traditional surrogate model in SMBO uses the hyperparameter $\boldsymbol{\lambda}^i$ as its input. In DOIT, the input $\text{VEC}^i$ also includes a vector representing the parameter state $\boldsymbol{\theta}^i$ to learn the impact under different parameter states. To reduce input size and cost, DOIT uses Principal Component Analysis (PCA) (Labrín & Urdinez, 2020) to compress the parameters layer by layer, lowering the training cost for the surrogate model. Additionally, DOIT selects only a subset of layers (e.g., the classifier layer with a few hidden layers) as inputs for the surrogate model to minimize training costs. In the case of PEFT, DOIT focuses solely on the trainable parameters (e.g., the matrices $A$ and $B$ in LoRA (Hu et al., 2021)).

**The output of the surrogate model.** In contrast to the results obtained from training to convergence, DOIT emphasizes the "short-term" benefits each optimizer can achieve given the current parameter state. Therefore, the output of the surrogate model does not use the final loss or accuracy, but instead employs a computed performance score $s \in [-1, 1]$. During a switch cycle, DOIT performs $\tau$ iterations of training, resulting in a set of losses, denoted as $\boldsymbol{l} = \{l_1, l_2, ..., l_\tau\}$, and $l_0$ represents the loss before training. DOIT first calculates the loss variation $\Delta l_i = \frac{l_{i-1} - l_i}{\max(l_i, l_{i-1})}$ for each iteration to get the average reduction $\mu_\Delta$ and variance $\sigma_\Delta$. To estimate the optimization performance of different configurations, DOIT combines the considerations of exploration (i.e., variance $\sigma_\Delta$) and exploitation (i.e., mean $\mu_\Delta$) to calculate a weighted score $s = \mu_\Delta + \alpha\sigma_\Delta$. However, such a weighted score

overlooks the direction of variance. For instance, in Figure 6, optimizers $o_1$ and $o_3$ have the same $\mu_\Delta$ and $\sigma_\Delta$, yet $o_3$ achieves a lower loss than $o_1$ during training. A similar issue arises in the comparison between $o_2$ with $o_1$ and $o_3$. To address this problem, we incorporate boundary considerations into the performance calculation, including the upper bound $\Delta_{\text{UPPER}} = \frac{l_0 - \max(\boldsymbol{l})}{\max(l_0, \max(\boldsymbol{l})) \times \tau}$ and lower bound $\Delta_{\text{LOWER}} = \frac{l_0 - \min(\boldsymbol{l})}{\max(l_0, \min(\boldsymbol{l})) \times \tau}$. Then, the final optimization performance score is defined as follows,

$$s = \tanh(\frac{1}{3}(\mu_\Delta + \Delta_{\text{UPPER}} + \Delta_{\text{LOWER}}) + \alpha\sigma_\Delta) \qquad (2)$$

where $\alpha$ represents the weight for variance.

**The initial weighted random selection.** To obtain enough training experience for the construction of surrogate models, DOIT trains with random configurations at the start of training. Though randomly selecting configurations for initial training can yield the necessary experience, DOIT employs a weighted random initialization method to enhance the performance of the initial training. Specifically, DOIT maintains a sampling weight $\boldsymbol{\omega}_r[j] \in [0, 1]$ for each type of optimizer $o_j$ and its surrogate model $g_j$, presenting the probability of being sampled. This sampling weight is initially assigned a value of 1 to achieve a random initialization. After completing the training with the current configuration, the corresponding optimizer's sampling weight $\omega_j$ is updated to the normalized performance score as shown in Equation (3), where $\omega_{\min}$ represents the minimal threshold.

$$\boldsymbol{\omega}_r[j] = \max(\frac{1}{2}(s+1), \omega_{\min}). \qquad (3)$$

## 3.4 Selecting Optimizers with Acquisition Function

Although the calculated optimization performance $s$ can be used to select configurations directly, given the volatility of the loss and the complexity of model training, DOIT considers additional factors in the design of acquisition function, including variance, transferability, and the training process. In this section, we introduce these considerations in DOIT's acquisition function.

**Consideration of variance.** Benefiting from the advantages of the Gaussian process model, the surrogate model can provide both the mean score $s_\mu$ and an estimate of the variance $s_\sigma$. Similar to traditional hyperparameter optimization methods, DOIT also incorporates a trade-off between exploration and exploitation in the acquisition function as,

$$\text{ACQ}(s_\mu, s_\sigma) = s_\mu + \alpha s_\sigma, \qquad (4)$$

where $\alpha$ represents the weight for variance, consistent with the definition in Equation (2).

**Consideration of transferability.** The training cost of DNNs is closely related to the initial model parameter state $\boldsymbol{\theta}^0$. In fine-tuning, closely related upstream and downstream tasks (i.e., high transferability between the pre-trained model and the new task) are easier to train than those that are dissimilar. Considering the idea that *"a pre-trained model with lower transferability necessitates more substantial tuning adjustments"*, we use the model's transferability $\omega_t$ as the weight of the variance in the acquisition function, as shown in the following equation,

$$\text{ACQ}(s_\mu, s_\sigma, \omega_t) = s_\mu + (1 - \omega_t)s_\sigma. \qquad (5)$$

The transferability $\omega_t$ is calculated using two types of evaluation metrics, including performance-based metric $\omega_p$ and distribution-based metrics $\omega_d$. First, the performance-based metric $\omega_p \in [0, 1]$ is the test result (e.g., accuracy) which is directly tested with the pre-trained model without deep refining. Second, the distribution-based metrics analyze the distribution of the output vectors or labels to estimate the model's transferability, including LogME (You et al., 2021) and Leep (Nguyen et al., 2020). Equation (6) presents the definition of transferability weight.

$$\omega_t = \beta\omega_p + (1 - \beta)\frac{1}{k}\sum_{i=1}^{k} \text{sigmoid}(\omega_d^i). \qquad (6)$$

where $\omega_d^i$ represents $k$ distribution-based metrics and $\beta$ the weight for two kinds of metrics. A sigmoid function constrains distribution-based metrics to $[0, 1]$, aligning them with performance-based metrics. The weighted sum then reflects the model's transferability to the current task: higher values indicate greater transferability, lower values the opposite.

**Consideration of training process.** DOIT takes into account the differing needs in early and later training stages, specifically that *"after the model becomes stable, smaller tuning adjustments are needed"*. As training progresses, the model gradually captures task knowledge, leading to a stabilization of the training loss. At the later stages of the training, the target position on the parameter surface is constrained within a smaller range. In this context, optimizers with larger amplitudes may disrupt the tuning process. Therefore, the proportion of variance in the acquisition function should be reduced. Hence, we introduce a periodic halving of the weight for variance information in DOIT as Equation (7), where $i$ represents the current iteration and $n$ represents the halving period.

$$e = \text{sigmoid}(s_\mu + (1 - 2^{-\lfloor i/n \rfloor} \cdot \omega_t)s_\sigma). \tag{7}$$

## 4 DISCUSSION

To further explore the characteristics of DOIT, we conduct a discussion in this section. And a further discussion about the memory overhead is presented in Appendix C.

### 4.1 ANALYZING THE DIFFERENCES BETWEEN DOIT AND HPO

**Compare with AutoML.** The optimizer can be viewed as a hyperparameter in DNNs, and automatically selecting it is a form of HPO and AutoML. However, vanilla training treats this as a static HPO problem, keeping the configuration fixed throughout. This static setting can be formulated as,

$$c^* = \arg\min_{c \in \mathcal{O} \times \Lambda \times \mathcal{T}} \mathcal{L}(\boldsymbol{\theta}^0, \mathcal{D}, c), \tag{8}$$

where $c = (o, \lambda, t)$ denotes the configurations (as defined in Section 3.2). Compared to DOIT's dynamic tuning (i.e., Equation 1), the static setting restricts the way model parameters are updated and the collaboration among different optimizers. In addition, though some hybrid methods (e.g., SWATS (Keskar & Socher, 2017)) have been proposed, they are still limited to simple transitions rather than dynamic adjustments among various optimizers based on training states.

**Compare with SMBO.** DOIT adopts the idea of surrogate models and the acquisition function in SMBO (e.g., Bayesian Optimization), but there are significant differences between them. While SMBO focuses solely on hyperparameter effects and performs static tuning, DOIT incorporates model parameter states and training progress to estimate "short-term" gains, enabling dynamic tuning. Unlike SMBO's goal of finding a fixed configuration, DOIT supports adaptive collaboration of optimizers throughout training to achieve better training performance.

### 4.2 ANALYZING THE ADVANTAGES OF DOIT

**Accuracy.** DOIT achieves a DNN training with interleaving optimizers, enabling collaboration among multiple optimizers. This dynamic optimizer tuning not only integrates the strategies of different optimizers but may also yield an optimization path (i.e., the final trained model) that a single optimizer cannot achieve, resulting in higher accuracy. Figure 7(a) provides examples across three different functions, illustrating that DOIT can discover optimization paths that a single optimizer cannot achieve. Similarly, the final model weights obtained from training for the same epochs on the *cifar10* dataset using different optimizers show significant differences, as illustrated in Figure 7(b). Our proposed DOIT, which employs multiple optimizers, expands the search space of traditional training, raising the accuracy upper bound.

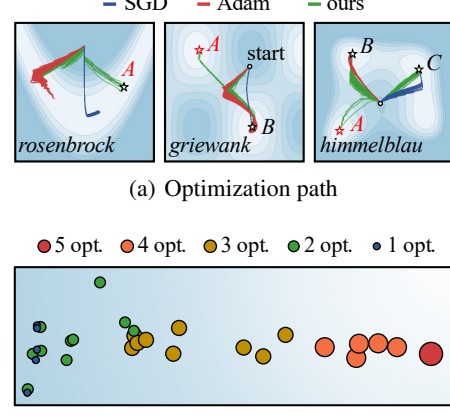

(a) Optimization path

(b) Trained model weights (PCA)

Figure 7: Analyze of interleaving training.

**Training efficiency and Memory Overhead.** To evaluate the time overhead of DOIT, we use ResNet18 on *cifar10* as an example. Table 1 summarizes the FLOPs of DOIT's main components, with detailed calculations provided in Appendix B. Under typical settings ($n_{\text{EPOCH}} = 100$, $n_{\text{BZ}} = 64$, $\tau = 20$, $m = 10^3$), the additional time introduced by DOIT is minimal (less than 1%). The

optimizer overhead also remains within the normal range of the candidate optimizers. We further discuss convergence behavior and memory cost in Appendix B-C.

Table 1: Time cost analyze with ResNet18.

|  | forward | gradient | optimizer | transferability | surrogate | acquisition |
|---|---|---|---|---|---|---|
| FLOPs | $2.2 \times 10^{16}$ | $4.4 \times 10^{16}$ | $0.2 \sim 2.0 \times 10^{13}$ | $2.2 \times 10^{13}$ | $5.3 \times 10^{14}$ | $1.0 \times 10^{14}$ |
| Proportion | 33.3% | 66.7% | quite small | 0.3‰ | 8.0‰ | 1.5‰ |

# 5 EXPERIMENTAL STUDY

To investigate the rationality of DOIT, we first present overall results across various models and tasks, followed by case studies analyzing its switching behavior and selection preferences. In addition, several independent experiments are presented to assess key designs in DOIT. The main experimental settings are summarized below, with more detailed descriptions in Appendix D.

**Datasets.** We evaluated DOIT on 6 CV classification datasets (i.e., *usps*, *mnist*, *stl10*, *cifar10*, *imagenet*, *imagenet-a*) and 2 NLP datasets (i.e., *mrpc* and *qqp*), along with 3 additional tasks: machine translation (*wmt14*), regression (*eunite*), and object detection (*coco*).

**Models.** We used 4 ImageNet pre-trained models available from PyTorch (i.e., ResNet18, ResNet152, MobileNet V2, and ViT) and a pre-trained NLP model from HuggingFace (i.e., RoBerta). For other tasks, we used OPUS, TabNet, and Faster R-CNN, respectively.

**Baselines.** We used 9 optimizers (i.e., SGD, SGDM, Adagrad, RMSProp, Adam, ASGD, AdamW, Nadam, and Adamax) and 4 hybrid optimizers (i.e., SWATS, Padam, AdaBound, and AGD). In addition, we also compared DOIT with optimizer using learning rate schedulers.

**Setup.** We set initial learning rates to $[0.1, 0.01, 0.001]$ and training epochs to 100. For DOIT, we used $n_{ini} = 50$, a switch cycle of $\tau = 25$ iterations, and an optimizer space of [SGD, SGDM, Adagrad, RMSprop, Adam]. Each experiment was repeated 5 times with different seeds on a Linux server with a 128-core 2.6GHz Intel Xeon Platinum 8358 CPU and 512GB RAM.

## 5.1 OVERALL PERFORMANCE OF DOIT

We first present the comparison results between DOIT and commonly used methods on classification tasks, including full training and PEFT. We then broadened the evaluation to additional models and tasks, further comparing DOIT with other approaches to demonstrate its effectiveness.

Table 2: Test accuracy (%) of full and PEFT training with ViT and RoBerta.

|  | full training | | | | | PEFT | | | |
|---|---|---|---|---|---|---|---|---|---|
|  | usps | mnist | stl10 | cifar10 | imagenet | usps | stl10 | mrpc | qqp |
| SGD | $97.46_{\pm 0.70}$ | $99.50_{\pm 0.04}$ | $97.83_{\pm 0.22}$ | $97.53_{\pm 0.03}$ | $78.73_{\pm 0.38}$ | $94.42_{\pm 0.21}$ | $97.75_{\pm 0.16}$ | $85.21_{\pm 0.35}$ | $82.13_{\pm 0.52}$ |
| SGDM | $97.68_{\pm 0.11}$ | $99.65_{\pm 0.04}$ | $96.61_{\pm 0.04}$ | $97.58_{\pm 0.04}$ | $78.36_{\pm 0.12}$ | $95.67_{\pm 0.14}$ | $98.37_{\pm 0.10}$ | $85.54_{\pm 0.69}$ | $83.30_{\pm 0.63}$ |
| Adagrad | $93.40_{\pm 0.04}$ | $98.24_{\pm 0.50}$ | $78.66_{\pm 2.54}$ | $60.95_{\pm 0.62}$ | $77.59_{\pm 0.03}$ | $95.37_{\pm 0.21}$ | $98.34_{\pm 0.09}$ | $84.94_{\pm 0.59}$ | $83.47_{\pm 0.79}$ |
| RMSprop | $95.25_{\pm 0.04}$ | $98.14_{\pm 0.09}$ | $88.62_{\pm 4.45}$ | $78.09_{\pm 0.92}$ | $74.01_{\pm 0.08}$ | $94.64_{\pm 0.53}$ | $97.91_{\pm 0.07}$ | $84.09_{\pm 0.76}$ | $82.09_{\pm 0.63}$ |
| Adam | $93.26_{\pm 0.78}$ | $99.01_{\pm 0.08}$ | $82.26_{\pm 1.16}$ | $75.23_{\pm 0.92}$ | $73.96_{\pm 0.11}$ | $94.47_{\pm 0.49}$ | $98.36_{\pm 0.03}$ | $86.52_{\pm 0.71}$ | $82.27_{\pm 0.71}$ |
| SWATS | $94.00_{\pm 1.23}$ | $98.73_{\pm 0.13}$ | $88.03_{\pm 0.44}$ | $66.15_{\pm 4.74}$ | $76.93_{\pm 0.19}$ | $95.12_{\pm 0.21}$ | $98.38_{\pm 0.10}$ | $86.27_{\pm 0.62}$ | $80.79_{\pm 0.81}$ |
| Padam | $97.58_{\pm 0.11}$ | $99.66_{\pm 0.04}$ | $90.81_{\pm 0.06}$ | $96.03_{\pm 0.03}$ | $77.47_{\pm 0.16}$ | $95.72_{\pm 0.42}$ | $98.38_{\pm 0.10}$ | $80.64_{\pm 0.32}$ | $73.04_{\pm 0.91}$ |
| AdaBound | $87.64_{\pm 0.84}$ | $97.54_{\pm 0.13}$ | $86.33_{\pm 2.39}$ | $70.91_{\pm 4.35}$ | $75.93_{\pm 0.13}$ | $95.42_{\pm 0.11}$ | $98.30_{\pm 0.14}$ | $68.38_{\pm 0.59}$ | $78.64_{\pm 0.49}$ |
| ours | $\mathbf{97.81_{\pm 0.21}}$ | $\mathbf{99.71_{\pm 0.02}}$ | $\mathbf{98.21_{\pm 0.19}}$ | $\mathbf{98.04_{\pm 0.03}}$ | $\mathbf{79.98_{\pm 0.01}}$ | $\mathbf{96.12_{\pm 0.10}}$ | $\mathbf{99.01_{\pm 0.09}}$ | $\mathbf{87.99_{\pm 0.13}}$ | $\mathbf{85.57_{\pm 0.14}}$ |

**Experiments on full training and PEFT.** We first compared DOIT with optimizers from its search space and hybrid methods on classification tasks, as shown in Table 2. For CV tasks, we used ViT (head fine-tuning (Poojary & Pai, 2019) for PEFT), and RoBerta with LoRA (Hu et al., 2021) for NLP. Experimental results show that DOIT often achieves higher accuracy (i.e., 1%-3% improvement) and lower variance. Meanwhile, existing hybrid methods perform poorly on some complex datasets (e.g., *qqp*), whereas DOIT adapts effectively to different datasets. To investigate the convergence of DOIT, we recorded the convergence time (i.e., the change of loss is less than $1 \times 10^{-4}$

in 10 consecutive epochs) and loss trajectories of each method, as shown in Figure 8 and Figure 9. The results show that DOIT achieves faster convergence in terms of time and iterations. Additional results are provided in Appendix E.

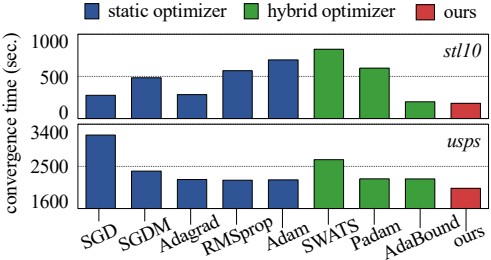

Figure 8: Convergence time on PEFT.

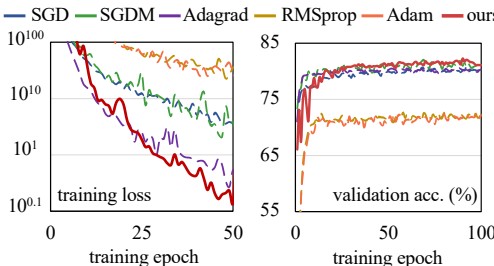

Figure 9: Loss and accuracy line on *cifar10*.

**More experiments.** In addition to the experiments above, we conducted further evaluations on a wider range of tasks and models, and compared DOIT against more baselines. (i) **More complex dataset**: we conducted experiments a complex imbalanced dataset *imagenet-a* (Hendrycks et al., 2021) as shown in Table 3. DOIT's dynamic adaptation to critical saddle points enhances performance on complex problems, resulting in over a 2% improvement in top-1 accuracy. (ii) **More models, tasks, and baselines**: Table 4 presents results on additional models, tasks, and baselines, where DOIT consistently achieves the best performance. More experimental results and detailed settings can be found in Appendix E. (iii) **Detailed time cost analyze:** Figure 10 illustrates the time consumption of different components across various methods. Compared to static optimizers, the additional time cost introduced by DOIT is minimal (less than 0.5%), which aligns with our analysis in Section 4.2.

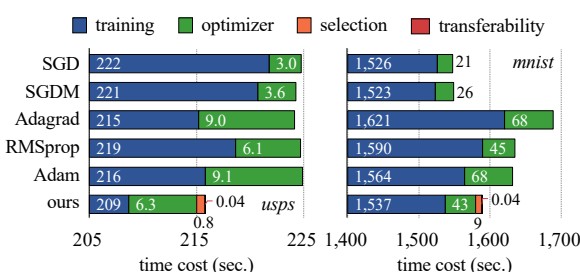

Figure 10: Time cost of each step.

Table 3: Test accuracy (%) of the full training on *imagenet-a*.

|        | SGD   | SGDM  | Adagrad | RMSprop | Adam  | SWATS | Padam | AdaBound | ours      |
|--------|-------|-------|---------|---------|-------|-------|-------|----------|-----------|
| acc@1  | 15.24 | 16.19 | 5.20    | 4.13    | 3.53  | 3.68  | 16.31 | 4.33     | **18.47** |
| acc@3  | 30.74 | 31.20 | 13.12   | 10.42   | 9.87  | 8.94  | 31.76 | 11.23    | **33.83** |
| acc@5  | 38.29 | 39.68 | 17.49   | 17.52   | 13.83 | 12.55 | 40.01 | 15.46    | **41.32** |

Table 4: Experiments with more models, tasks, and baselines.

(a) more models

|          | stl10 test accuracy (%) | | |
|----------|----------|-----------|-----------|
|          | ResNet18 | ResNet152 | MobileNet |
| SGDM     | 91.87    | 96.66     | 92.14     |
| Adam     | 91.02    | 96.11     | 91.31     |
| SWATS    | 92.02    | 96.66     | 92.77     |
| Padam    | 92.36    | 96.51     | 91.69     |
| AdaBound | 92.23    | 95.44     | 91.39     |
| ours     | **93.56** | **97.60** | **93.28** |

(b) more tasks

| metric model | wmt14 BLEU OPUS | eunite R2 TabNet | coco AP50 RCNN |
|--------------|-------|-------|-------|
| SGD          | 27.91 | 71.81 | 57.44 |
| SGDM         | 28.12 | 70.32 | 57.26 |
| Adagrad      | 27.14 | 71.22 | 56.46 |
| RMSprop      | 27.41 | 67.88 | 53.08 |
| Adam         | 28.24 | 69.83 | 56.46 |
| ours         | **28.68** | **73.12** | **59.21** |

(c) more baselines

|        | usps  | mnist |
|--------|-------|-------|
| ASGD   | 95.83 | 99.25 |
| AdamW  | 95.62 | 98.97 |
| Nadam  | 95.32 | 99.18 |
| Adamax | 96.06 | 99.32 |
| AGD    | 95.96 | 99.19 |
| stepLR | 91.31 | 94.72 |
| cosLR  | 95.80 | 99.46 |
| ours   | **96.81** | **99.51** |

## 5.2 ANALYZE THE SWITCHING PROCESS OF DOIT WITH CASE STUDIES

To further investigate the patterns and preferences of DOIT in optimizer selection, we conducted several case studies. (i) **When does the optimizer switching occur?** The left plot of Figure 11 shows the training process of DOIT on a simple dataset, *hymenoptera*, where it switches only between SGD and SGDM. After random initialization, DOIT performs switching operations when a decrease in the convergence speed of the optimizer (Point *A*) or detects a local stable state (Point *C*). In addition, during tuning, DOIT may also undergo temporary switches to adjust the optimization state (Point *B*). (ii) **What is DOIT's selection preference like?** The right plot of Figure 11 presents DOIT's switching probabilities and selection patterns across different training stages (with switching limited to SGD and Adam). In summary, we have identified the following characteristics:

- **"As training progresses, the selection of optimizers becomes more stable (fewer switches)."** Figure 11(b) illustrates the probability of optimizer changes at different training stages. It shows that the likelihood of switching optimizers gradually decreases over time.

- **"In the early training phase, there is a tendency to select optimizers that converge faster (e.g., Adam), while in the later phase, DOIT tends to select more stable optimizers (e.g., SGD)."** As shown in subplots (c) and (d), DOIT demonstrates a stronger preference for SGD in the later training stages, which aligns with the findings of SWATS (Keskar & Socher, 2017).

- **"DOIT's selection preference is more apparent in a large dataset."** Comparing subplots (c) and (d), DOIT shows clearer selection preferences on *mnist* than *usps*. Larger datasets provide more experience for surrogate model training, allowing for more stable selection of specific optimizers.

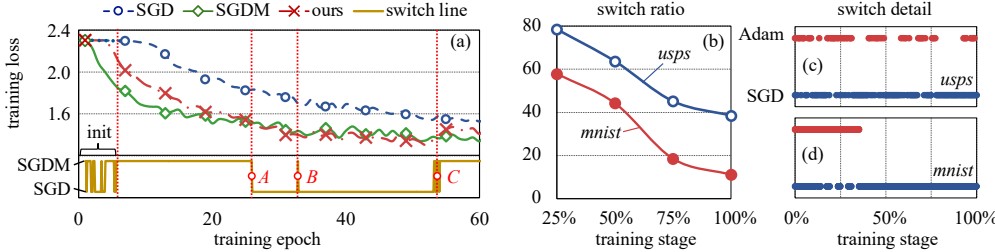

Figure 11: The training loss line of the case study.

## 5.3 INDEPENDENT EXPERIMENTS

Moreover, we conducted independent experiments to further analyze the effectiveness of DOIT. We outline the main conclusions, with further details available in Appendix F. (i) **Varying search space.** Figure 12 shows the impact of expanding the search space (i.e., more optimizer types and hyperparameter options) on DOIT's performance. It demonstrates that DOIT continues to perform well in the enlarged search space, with only a minimal increase in time cost. (ii) **Ablation experiment of key designs.** To assess the effectiveness of DOIT's key design components, we conducted a series of ablation studies. These include comparisons on the optimizer selection strategy (DOIT vs. random switching and periodic replacement), acquisition function design (DOIT vs. versions without transferability weight $\omega_t$ or training process halving), initial selection strategy (DOIT's weighted random vs. uniform random), and parameter compression method (PCA

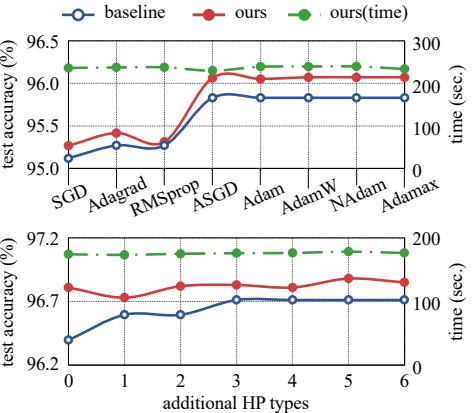

Figure 12: Results with varying space.

used in DOIT vs. random projection and UMAP). Experimental results demonstrate that each component of DOIT's design contributes to improving both convergence speed and model quality. (iii)

**Hyperparameter analysis.** We also performed an experimental analysis on the setting of hyperparameters in DOIT, including the initial step $n_{ini}$, switch step size $\tau$, number of PCA components, and selected compression layers. The results show that a small initial step (e.g., only 10 for small dataset *usps*), switch step size (10% of an epoch) and PCA components (e.g., 2) can help achieving good accuracy with a limited time cost.

## 6 CONCLUSION

Optimizers play a pivotal role in training deep neural networks. However, current training approaches (i.e., using a static optimizer or simple hybrid methods) struggle to adapt to the evolving optimization landscape across different training states. This lack of adaptability can degrade both convergence and model quality. To address this limitation, we propose a novel method called Dynamic Optimizer Interleaving Training (DOIT) method, which dynamically selects suitable optimizers based on the training states. During training, DOIT leverages surrogate models to estimate the performance of various optimizers given the current parameter state. It then combines these estimates with transferability assessments and processes information through a special acquisition function to predict potential optimization gains. At each training state, DOIT selects the optimizer with the highest score for training. Experiments results show that DOIT achieves higher accuracy (i.e., 1% to 3% improvement) and faster convergence (i.e., 2% to 10% faster).

**Limitations and future work** This work focuses on dynamic optimizer switching during training, particularly regarding the types of optimizers used. Our method does not consider the inheritance of internal states (e.g., momentum) when switching optimizers. While prior studies (Chen et al., 2018; Yue et al., 2023) explored inheritance between two optimizers, extending it to many candidates greatly increases complexity. As discussed in Appendix G, simple inheritance does not yield consistent gains and may introduce instability, so robust cross-optimizer state transfer remains an important direction for future work.

## ACKNOWLEDGMENTS

This work is supported by the National Natural Science Foundation of China (Grant No. 62376099) and the Natural Science Foundation of Guangdong Province (Grant No. 2024A1515010989).

## REPRODUCIBILITY STATEMENT

All models, datasets, and baseline methods used in this work are publicly available. Detailed descriptions of experimental settings, hyperparameter configurations, and training procedures are provided in the Experiments section and further elaborated in the Appendix. To ensure full reproducibility, we will also release the complete implementation and scripts for all experiments.

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

## A  OVERALL ALGORITHM AND PSEUDOCODE OF DOIT

For ease of reading, we provide a detailed explanation of key notations used in this paper for DOIT in Table 5.

Table 5: The description of notations used in DOIT.

| notation | description |
|---|---|
| switch step $\tau$ | the number of iterations for a single switch cycle. |
| init step $n_{ini}$ | the number of switch cycles for the initial phase. |
| selection weights $\omega_{\mathbf{r}}$ | the sample weights for the initial phase |
| transferability weight $\omega_t$ | transferability weight which is calculated by the performance-based metric $\omega_t$ and distribution-based metric $\omega_d$ |

In this section, we provide an overview of DOIT with the pseudocode in Algorithm 1. For ease of understanding, we only switch the types of optimizers in the description while keeping the hyperparameters and training time constant (i.e., $t^i = \tau$ iterations). Before the model's training, DOIT first calculates the transferability weight $\omega_t$ to assist in the subsequent selection of optimizers (Line 2). Then, a randomly selected initial optimizer $o_j$ is used for model training and loss calculation (Lines 5-6). When the number of iterations meets the time constant $\tau$, DOIT calculates the performance score $s$ based on all losses within switch cycle $losses$ (Line 8). Subsequently, the current optimizer $o_j$ is updated by the surrogate model $g_j$ using the optimization gain $s$ and the model features $v$ calculated at the end of the previous round (Lines 9-10). Based on the results from weighted random selection or surrogate model selection, DOIT obtains the suitable optimizer for the next training stage (Lines 10-15) and continues this process iteratively until the final training results $\theta^n$ are achieved.

For the selection of the suitable optimizer, DOIT employs two types of methods: the weighted random selection and the surrogate model selection for the following steps. In the initial steps (i.e., $n_t < \tau$), DOIT uses the optimization gain $s$ to update the sampling weight $\boldsymbol{\omega}_r$ for randomly select configurations (Lines 10-13). After obtaining enough training results (i.e., $n_t \geq \tau$), DOIT utilizes the trained surrogate models to select the configurations used for the following training (Lines 13-15). The configuration with the highest score (i.e., Equation 7) is selected for the next training iteration.

---

**Algorithm 1:** Basic framework of our proposed DOIT.

---

**Input:** Training Dataset $D = \{D_{train}, D_{test}\}$, $m$ optimizers $\mathcal{O} = \{o_1, ..., o_m\}$, the model for training $M$ with initial parameter state $\boldsymbol{\theta}^0$, the init steps $n_{ini}$, the switch step size $\tau$, the training epochs $n_{epoch}$.

1 **initialize:** $n_t \leftarrow 0, n \leftarrow 0, \mathcal{G} \leftarrow \{g_1, g_2, ..., g_m\}, \boldsymbol{\omega}_r \leftarrow \{1|1, ..., m\}, j \leftarrow \text{RandomSelect}(m)$,
  $losses \leftarrow [], \boldsymbol{v} \leftarrow \text{CompressModel}(M, \boldsymbol{\theta}^0)$
  /* calculate the transferability weight before the training      */
2 $\boldsymbol{\omega}_t \leftarrow \text{CalculateTransferabilityWeight}(M, \boldsymbol{\theta}^0, D_{train})$    (as Equation 6)
  /* performance evaluation and optimizer switch during FT      */
3 **for** $i$ *in* $[1, n_{epoch}]$ **do**
4    **for** BATCH *in* $D_{train}$ **do**
5       $n \leftarrow n + 1, \boldsymbol{\theta}^n \leftarrow \text{TrainModel}(M, \boldsymbol{\theta}^{n-1}, \text{BATCH}, o_j)$
6       $l \leftarrow \text{CalculateLoss}(M, \boldsymbol{\theta}^n, \text{BATCH}), losses.\text{append}(l)$
7       **if** $n\%\tau = 0$ **then**
8          $s \leftarrow \text{CalculateOptimizationGain}(losses)$    (as Equation 2)
9          $g_j \leftarrow \text{UpdateSurrogateModel}(g_j, \boldsymbol{v}, s)$
10          $\boldsymbol{v} \leftarrow \text{CompressModel}(M, \boldsymbol{\theta}^n)$
           // init steps with a weighted random selection
11          **if** $t < n_{in}$ **then**
12             $\boldsymbol{\omega}_r[j] \leftarrow \text{UpdateSampleWeight}(s)$    (as Equation 3)
13             $j \leftarrow \text{WeightedRandomSelect}(\boldsymbol{\omega}_r)$
           // following steps with a surrogated selection
14          **else**
15             $j \leftarrow \text{SurrogatedSelect}(\boldsymbol{v}, \omega_t, \mathcal{G}, \mathcal{O})$
16          $n_t \leftarrow n_t + 1, losses \leftarrow []$
17    **end**
18 **end**
**Output:** the trained model $M$ with parameter state $\boldsymbol{\theta}^n$

---

## B  DETAILS ABOUT THE ANALYSIS OF TRAINING EFFICIENCY

Here, we provide a detailed description of the time cost calculation mentioned in Section 4.2, based on the following analysis scenario:

- **Model:** ResNet18, who contains approximately $W = 1.17 \times 10^7$ parameters. Its forward pass requires around $t_{\text{FW}} \approx 3.66 \times 10^9$ FLOPs[1], and we estimate the cost of backpropagation and gradient updates as twice that of the forward pass, yielding $t_{\text{BP}} \approx 7.32 \times 10^9$ FLOPs.
- **Dataset:** *cifar10*, whose feature dimensions $D \approx 3 \times 10^3$, instance number $N = 6 \times 10^4$, and class number $K = 10$.
- **Hyperparameter setting:** We set the epoch number $n_{\text{EPOCH}} = 100$, batch size $n_{\text{BZ}} = 64$, switching iteration number $\tau = 20$, and the optimizer search space of DOIT $m = 10^3$.

For vanilla training, the total time cost primarily consists of three components: forward propagation $t_{\text{forward}}$, backward propagation $t_{\text{backward}}$, and parameter updates $t_{\text{update}}$. The FLOPs of forward and backward propagation are related to the number of instances and training epochs, and are given as follows,

$$t_{\text{forward}} = t_{\text{FW}} \times N \times n_{\text{EPOCH}} \approx 2.2 \times 10^{16},$$

$$t_{\text{backward}} = t_{\text{BP}} \times N \times n_{\text{EPOCH}} \approx 4.4 \times 10^{16}.$$

---

[1] https://paddleclas.readthedocs.io/en/latest/models/ResNet_and_vd_en.html

In addition, the FLOPs for parameter updates are dependent on the optimizer type, the number of model parameters $W$, and the batch size $n_{\text{BZ}}$. Different optimizers require different computational steps $n_{\text{opt}}$ for parameter updates. For example, in SGD, each parameter update requires 2 computations, whereas Adam requires approximately 18 computations. Using these two optimizers as examples, the FLOPs for parameter updates are as follows:

$$t_{\text{update}} = n_{\text{opt}} \times W \times \frac{N}{n_{\text{BZ}}} \times n_{\text{EPOCH}} \approx 2.2 \times 10^{12} \sim 2.0 \times 10^{13}.$$

For DOIT, the FLOPs for forward $t'_{\text{forward}}$ and backward propagation $t'_{\text{backward}}$ are the same as those in vanilla training, while the parameter update FLOPs $t'_{\text{update}}$ is in the range of the value in the optimizer space. Thus, we have,

$$t'_{\text{forward}} = t_{\text{forward}} \approx 2.2 \times 10^{16}, \quad t'_{\text{backward}} = t_{\text{backward}} \approx 4.4 \times 10^{16},$$
$$t'_{\text{update}} \in [\max(t_{\text{update}}), \min(t_{\text{update}})] \approx [0.2 \times 10^{13}, 2 \times 10^{13}].$$

Moreover, compared to vanilla training, DOIT mainly introduces additional time overhead from the following three components,

- **transferbility assessment:** The assessment includes the computation for two distribution-based metrics (i.e., LEEP and LogME) and one performance-based metric. Among them, the time cost for the LEEP and performance-based metrics is equivalent to a single forward pass (Nguyen et al., 2020), while the computational complexity of LogME is $t_{\text{logme}} = KD^2 + NKD + D^3 + ND^2 \approx 3 \times 10^{10}$ (You et al., 2021). Since the transferability evaluation is performed only once before training, its FLOPs is equal to,

$$t'_{\text{transfer}} = t_{\text{FW}} \times N + t_{\text{logme}} \approx 2.2 \times 10^{13}.$$

- **surrogate model updating:** This time consumption includes the PCA compression of the selected parameters and the updating of the Gaussian process model (primarily driven by the computation of the covariance matrix). The time complexity of compression and updating is $O(W_{se}^2 D')$ and $O(n_{ob}^3)$, where $W_{se} \approx 10^4$ represents the number of selected parameters (i.e., only the last layer), $D' \approx 100$ represents the number of PCA components, $N_{sw} = n_{\text{EPOCH}} \frac{N}{\tau \times n_{\text{BZ}}} \approx 4.7 \times 10^3$ represents the total switching operations in the tuning process, and $n_{ob}$ is the number of observation points in surrogate model. In our computation, we set $n_{ob} = N_{sw}$, although the actual value is smaller due to the use of multiple surrogate models for different optimizers and varying update times. Then we can calculate the FLOPS as follows,

$$t'_{\text{surrogate}} = N_{sw} \times (W_{se}^2 D' + n_{ob}^3) \approx 5.3 \times 10^{14}.$$

- **acquisition function calculation:** The computation of the acquisition function is mainly determined by the performance evaluation of the surrogate model. Specifically, the prediction complexity of Gaussian processes for a single candidate configuration is $O(n_{ob}^2)$. Then the FLOPs of acquisition function calculation is,

$$t'_{\text{acquisition}} = N_{sw} \times m \times n_{ob}^2 \approx 1.0 \times 10^{14}.$$

Based on the above analysis, the additional FLOPs introduced by DOIT are quite small compared to vanilla training (less than 1%). While factors such as hardware, GPU parallelism, and optimizer type can influence the exact runtime, the overall increase in training time is consistent with the FLOPs conclusion. Moreover, due to careful optimizer selection, DOIT often achieves faster convergence.

As for the convergence behavior, although DOIT performs optimizer switching during training, its overall convergence behavior is guaranteed by the underlying optimizers. DOIT does not modify any optimizer's update rule and instead selects from standard optimizers at different stages, so the convergence behavior aligns with that of the chosen optimizer at each stage. Moreover, the acquisition function encourages DOIT to favor more stable optimizers in the later phases of training (as shown in Sec. 5.2), ensuring a stable and reliable convergence.

## C   MEMORY OVERHEAD ANALYSIS

We analyze the additional memory overhead introduced by DOIT to address concerns regarding optimizer states and surrogate models. Overall, the overhead is small, approximately 13% of a

ResNet18 parameter size, and remains consistent across models and datasets. The additional memory overhead in DOIT mainly comes from: (1) storing the history of compressed parameter vectors and performance scores ($M_{rec}$), and (2) maintaining Guassian process surrogate models ($M_{sur}$). Following the analysis settings in Appendix B, with switch steps $N_{sw} \approx 2.7 \times 10^3$ and PCA dimension $D' = 100$, the history cost is $M_{rec} = N_{sw} \times D' = 2.7 \times 10^5$. For $K = 5$ candidate optimizers, each surrogate model stores approximately $\frac{1}{K} N_{sw}$ points, so kernel memory costs $M_{sur} = K \times (\frac{1}{K} N_{sw})^2 \approx 1.46 \times 10^6$. Together, these account for roughly 12.5% of ResNet18's parameter size (less than 3% of total training), and are constant across model sizes and datasets, making the memory overhead acceptable in practice. In addition, regarding optimizer states, DOIT does not maintain states for multiple optimizers simultaneously. Instead, optimizer states are reset at each switch, so only one optimizer's state is active at any time, resulting in memory usage comparable to a single optimizer.

## D  DATASETS, MODELS, BASELINES, AND SETTINGS IN EXPERIMENTS

In this section, we provide a detailed description of the datasets, models, baselines, and experimental settings used in our experiments.

**Datasets.** We used 6 CV classification datasets (i.e., *usps*, *mnist*, *stl10*, *cifar10*, *imagenet*, *imagenet-a*) and 5 NLP datasets (i.e., *mrpc*, *rte*, *qqp*, *qnli*, and *mnli*), along with 3 additional tasks: machine translation (*wmt14*), regression (*eunite*), and object detection (*coco*). All the datasets were loaded from PyTorch (Paszke et al., 2019), LibSVM (Chang & Lin, 2011) or HuggingFace (Wolf et al., 2020).

- *usps* (Hull, 1994) (from PyTorch): a classical digit dataset automatically scanned from envelopes by the U.S. Postal Service containing a total of 9,298 16×16 pixel grayscale samples, which includes 10 classes of figures.

- *mnist* (LeCun et al., 1998) (from PyTorch): a handwritten digits dataset with 28x28 grayscale figures, which has a training set of 60,000 examples and a test set of 10,000 examples.

- *stl10* (Coates et al., 2011) (from PyTorch): a 10-classes 96x96 color figure dataset, which has 500 training images and 800 test images per class. The dataset is inspired by the *cifar-10* (Krizhevsky et al., 2009) but with some modifications.

- *cifar10* (Krizhevsky et al., 2009) (from PyTorch): a 10-classes 32x32 color figure dataset, which has 5,000 training images and 1,000 test images per class.

- *imagenet* (Russakovsky et al., 2015) (from PyTorch): The ImageNet 2012 dataset (also known as ILSVRC 2012) classification subset contains 1,000 categories. The training set is approximately 137 GB, while the validation set is around 6 GB. Each class in the training set contains about 1,300 images, totaling approximately 1.3 million images. Each class in the validation set contains 50 images, for a total of 50,000 images.

- *imagenet-a* (Hendrycks et al., 2021) (from HuggingFace): includes real-world, unaltered, and naturally occurring examples that were misclassified by the ResNet model. The dataset consists of 7,500 adversarially filtered images, which can severely impact the performance of machine learning models.

- *mrpc* (Dolan & Brockett, 2005) (from HuggingFace): Microsoft Research Paraphrase Corpus, which consists of 5.8k sentence pairs that were automatically extracted from online news sources. The sentence pairs have been annotated by human raters to indicate whether the sentences within each pair are semantically equivalent.

- *rte* (Wang et al., 2018) (from HuggingFace): the recognizing textual entailment datasets come from a series of annual textual entailment challenges. Examples are constructed based on news and Wikipedia text. The authors of the benchmark convert all datasets to a two-class split, where for three-class datasets, they collapse neutral and contradiction into not entailment, for consistency.

- *qqp* (Quora) (from HuggingFace): the Quora Question Pairs dataset, which consists of over 400,000 pairs of questions. Each question pair is annotated with a binary value indicating whether the two questions are paraphrases of each other.

- *qnli* (Demszky et al., 2018) (from HuggingFace): Stanford Question Answering Dataset, which is a question-answering dataset consisting of question-paragraph pairs, where one of the sentences in

the paragraph (drawn from Wikipedia) contains the answer to the corresponding question (written by an annotator).

- *mnli* (Williams et al., 2018) (Multi-Genre Natural Language Inference Corpus, a crowdsourced collection of sentence pairs with textual entailment annotations.
- *wmt14* (en-de) (Bojar et al., 2014) (from HuggingFace): is a collection of datasets used in shared tasks of the Ninth Workshop on Statistical Machine Translation. We used the German-English translation dataset, which contains 4.51 million training samples, 3,000 validation samples, and 3,000 test samples.
- *eunite* (Chen et al., 2004) (from LibSVM): the regression dataset for electricity load forecasting.
- *coco* (Lin et al., 2014) (from HuggingFace): the object detection datasets COCO2017, which contain 135K images for training and 5K images for validation.

**Models.** For the pre-trained models, we used 4 ImageNet pre-trained models available from Py-Torch (Paszke et al., 2019) (i.e., ResNet18, ResNet152, MobileNet V2, and ViT) and 4 pre-trained NLP models (i.e., RoBerta, GPT-2, QWen2-0.5B, and LLaMA-3.2-1B) from HuggingFace (Wolf et al., 2020) which trained on 124M tweets from January 2018 to December 2021, and finetuned for sentiment analysis with the TweetEval benchmark (Barbieri et al., 2020). It can be found that among the downstream tasks, *stl10*, *mrpc* and *qqp* are relatively close to the upstream task, and *usps* and *mnist* have a certain correlation with the upstream task. We selected downstream datasets with varying degrees of relevance to the upstream task, to better analyze the performance of the proposed method in different scenarios. For other tasks, we used OPUS (Tiedemann & Thottingal, 2020) from HuggingFace, TabNet (Arik & Pfister, 2020) from pytorch-tabnet library, and Faster R-CNN (Ren et al., 2016) from PyTorch, respectively.

**Baselines.** In the overall experiment, we compared 9 single optimizer methods (i.e., SGD (Robbins & Monro, 1951), SGDM (Sutskever et al., 2013), Adagrad (Duchi et al., 2011), RMSProp (Graves, 2013), Adam (Kingma & Ba, 2014), ASGD (Polyak & Juditsky, 1992), AdamW (Loshchilov & Hutter, 2019), Nadam (Dozat), and Adamax (Kingma & Ba, 2017)), 4 hybrid methods (i.e., SWATS (Keskar & Socher, 2017), Padam (Chen et al., 2018), AdaBound (Luo et al., 2019), and AGD (Yue et al., 2023)), and the proposed DOIT. The single methods are all from the PyTorch implementation, and except for the learning rate being set in [0.1,0.01,0.001] and the epoch number being set to 100, the other hyperparameters are set to the default values in PyTorch. Additionally, for the hybrid methods, we installed and used the original implementations from the authors via Github and PyPI. The hyperparameter settings were kept at their defaults, except for the epoch number, which was adjusted to be consistent with the other methods. For DOIT, we used $n_{ini} = 50$, a switch cycle of $\tau = 25$ iterations, variance weight $\alpha = 0.1$, weight for transferability assessment $\beta = 0.8$, halving period $n = n_{\text{EPOCH}}/4$, and an optimizer space of [SGD, SGDM, Adagrad, RMSprop, Adam]. In addition, due to the large size of the *imagenet*, we set the number of epochs to 10 and the batch size to 256. As for the Gaussian surrogate model, we used the GaussianProcessRegressor from scikit-learn (Pedregosa et al., 2011) with its default settings.

# E   MORE DETAILS AND RESULTS ABOUT OVERALL EXPERIMENTS

In this section, we provide additional experimental details and results in Section 5.1.

**Overall performance.** In addition to the test accuracy results of the ViT and RoBerta models on the classification dataset shown in Table 2, we also performed full training on ResNet18 and recorded the F1 score of RoBerta PEFT. The results are shown in Table 6.

**Convergence analysis.** In addition to the loss lines for full training *cifar10* in Figure 9, we also recorded the results on *usps* and *stl10*, as shown in Figure 13. For ease of observation, the baselines with significant fluctuations are not displayed in the figure. It can be observed that DOIT shows a faster convergence speed compared to the vanilla method.

**More models, tasks, and baselines.** Table 4(a) presents experimental results with more CV models under head fine-tuning. We provide extended results in Table 7. In the experiments, all methods have run 100 epochs without early stopping. In addition, Table 4(c) reports only the best results obtained with training using a scheduler, specifically those achieved with SGDM. Moreover, we further evaluate DOIT on additional NLP models and tasks. Specifically, the experiment includes

Table 6: Test accuracy (%) of full training ResNet18 and F1 score (%) of PEFT training RoBerta.

| | full training | | | | PEFT | | | |
| | usps | mnist | stl10 | cifar10 | usps | mnist | mrpc | qqp |
|---|---|---|---|---|---|---|---|---|
| SGD | $96.10_{\pm0.24}$ | $99.29_{\pm0.03}$ | $86.94_{\pm0.49}$ | $80.73_{\pm0.43}$ | $94.42_{\pm0.21}$ | $97.75_{\pm0.16}$ | $87.21_{\pm0.32}$ | $75.19_{\pm0.82}$ |
| SGDM | $95.83_{\pm0.30}$ | $99.47_{\pm0.05}$ | $86.99_{\pm0.36}$ | $81.64_{\pm0.60}$ | $95.67_{\pm0.14}$ | $98.37_{\pm0.10}$ | $86.27_{\pm0.41}$ | $75.27_{\pm0.82}$ |
| Adagrad | $96.00_{\pm0.94}$ | $99.40_{\pm0.06}$ | $83.38_{\pm7.99}$ | $80.57_{\pm0.14}$ | $95.37_{\pm0.21}$ | $98.34_{\pm0.09}$ | $87.29_{\pm0.46}$ | $73.28_{\pm0.83}$ |
| RMSprop | $95.30_{\pm0.55}$ | $99.13_{\pm0.17}$ | $69.91_{\pm2.06}$ | $71.92_{\pm0.41}$ | $94.64_{\pm0.53}$ | $97.91_{\pm0.07}$ | $89.24_{\pm0.74}$ | $74.09_{\pm0.92}$ |
| Adam | $95.13_{\pm0.53}$ | $99.11_{\pm0.06}$ | $76.49_{\pm2.04}$ | $72.33_{\pm0.84}$ | $94.47_{\pm0.49}$ | $98.36_{\pm0.03}$ | $90.37_{\pm0.92}$ | $74.92_{\pm0.84}$ |
| SWATS | $95.53_{\pm0.71}$ | $99.17_{\pm0.11}$ | $79.76_{\pm2.11}$ | $75.17_{\pm0.21}$ | $95.12_{\pm0.21}$ | $98.38_{\pm0.10}$ | $90.34_{\pm0.42}$ | $74.80_{\pm0.19}$ |
| Padam | $96.10_{\pm0.15}$ | $99.46_{\pm0.02}$ | $85.64_{\pm0.48}$ | $81.58_{\pm0.38}$ | $95.72_{\pm0.42}$ | $98.38_{\pm0.10}$ | $87.07_{\pm0.82}$ | $80.93_{\pm0.83}$ |
| AdaBound | $95.02_{\pm0.17}$ | $99.25_{\pm0.06}$ | $84.48_{\pm0.42}$ | $69.27_{\pm5.02}$ | $95.42_{\pm0.11}$ | $98.30_{\pm0.14}$ | $81.22_{\pm0.94}$ | $79.01_{\pm0.42}$ |
| ours | $\mathbf{96.81_{\pm0.21}}$ | $\mathbf{99.51_{\pm0.01}}$ | $\mathbf{88.23_{\pm0.23}}$ | $\mathbf{84.14_{\pm0.11}}$ | $\mathbf{96.12_{\pm0.10}}$ | $\mathbf{99.01_{\pm0.09}}$ | $\mathbf{91.36_{\pm0.15}}$ | $\mathbf{81.18_{\pm0.31}}$ |

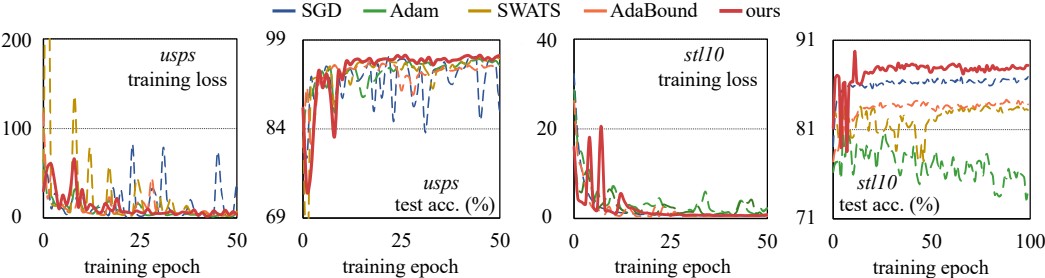

Figure 13: The training loss and test accuracy line graph.

3 pretrained models (GPT-2, QWen2-0.5B, and LLaMA-3.2-1B) and 5 GLUE datasets (mrpc, rte, qqp, qnli, and mnli). All models are fine-tuned using LoRA with a rank set to 512 for 5 epochs. Since Adam is the most commonly used and stable optimizer in NLP fine-tuning, Table 8 reports comparisons primarily against Adam. As shown in Table 8, DOIT consistently achieves higher test accuracy across all tasks and exhibits faster convergence. In terms of peak memory usage, although some differences remain due to memory-release behaviors, the overall additional overhead stays below 1%, which aligns with our analysis in Appendix C.

Table 7: The testAcc. and time for the vanilla method and our DOIT under the head fine-tuning.

| model | task method | usps testAcc. (%) | time (sec.) | mnist testAcc. (%) | time (sec.) | stl10 testAcc. (%) | time (sec.) |
|---|---|---|---|---|---|---|---|
| ResNet18 | SGDM | $68.86_{\pm0.71}$ | 534 | $71.59_{\pm1.88}$ | 4443 | $91.87_{\pm1.27}$ | 18551 |
| | Adam | $66.04_{\pm0.11}$ | 543 | $71.16_{\pm1.46}$ | 4876 | $91.02_{\pm1.87}$ | 14909 |
| | SWATS | $68.55_{\pm0.51}$ | 453 | $73.27_{\pm0.38}$ | 4651 | $92.02_{\pm1.86}$ | 14285 |
| | Padam | $65.32_{\pm1.20}$ | 505 | $70.41_{\pm0.52}$ | 4743 | $92.36_{\pm0.36}$ | 19919 |
| | AdaBound | $67.80_{\pm0.27}$ | 568 | $72.66_{\pm0.31}$ | 4944 | $92.23_{\pm1.37}$ | 34858 |
| | ours | $\mathbf{70.64_{\pm0.80}}$ | 531 | $\mathbf{74.79_{\pm3.38}}$ | 4807 | $\mathbf{93.56_{\pm0.41}}$ | 13265 |
| ResNet152 | SGDM | $72.27_{\pm0.12}$ | 4443 | $77.54_{\pm1.02}$ | 7437 | $96.66_{\pm0.32}$ | 78232 |
| | Adam | $71.73_{\pm1.26}$ | 4876 | $77.64_{\pm1.72}$ | 7636 | $96.11_{\pm0.32}$ | 48541 |
| | SWATS | $71.82_{\pm0.27}$ | 5121 | $78.47_{\pm0.18}$ | 9211 | $96.66_{\pm0.51}$ | 115560 |
| | Padam | $72.31_{\pm0.69}$ | 5594 | $76.18_{\pm1.17}$ | 8878 | $96.51_{\pm0.39}$ | 107856 |
| | AdaBound | $72.94_{\pm1.33}$ | 4508 | $78.33_{\pm0.67}$ | 12934 | $95.44_{\pm2.47}$ | 152592 |
| | ours | $\mathbf{74.36_{\pm0.41}}$ | 5007 | $\mathbf{79.60_{\pm0.60}}$ | 7645 | $\mathbf{97.60_{\pm0.28}}$ | 44449 |
| MobileNet v2 | SGDM | $91.28_{\pm0.39}$ | 18551 | $93.74_{\pm0.77}$ | 50294 | $92.14_{\pm0.71}$ | 11464 |
| | Adam | $89.86_{\pm0.22}$ | 12237 | $93.82_{\pm0.11}$ | 55024 | $91.31_{\pm0.71}$ | 15208 |
| | SWATS | $90.68_{\pm0.25}$ | 17921 | $93.90_{\pm0.28}$ | 57655 | $92.77_{\pm0.27}$ | 12971 |
| | Padam | $90.69_{\pm0.67}$ | 18940 | $92.71_{\pm0.22}$ | 51656 | $91.69_{\pm0.29}$ | 12941 |
| | AdaBound | $91.36_{\pm0.37}$ | 17328 | $93.67_{\pm0.94}$ | 46955 | $91.39_{\pm0.45}$ | 24367 |
| | ours | $\mathbf{92.32_{\pm0.39}}$ | 13248 | $\mathbf{94.65_{\pm0.06}}$ | 56132 | $\mathbf{93.28_{\pm0.79}}$ | 15042 |

Table 8: Test accuracy (%), convergence time (sec.), and peak memory (GB) for Adam and our DOIT across more NLP tasks with the LoRA fine-tuning.

| model | metric | method | mrpc | rte | qqp | qnli | mnli |
|-------|--------|--------|------|-----|-----|------|------|
| GPT2 | acc. (%) | Adam | 71.43 | 58.35 | 86.02 | 82.78 | 78.08 |
| | | ours | 72.23 | 59.13 | 87.48 | 83.01 | 79.14 |
| | time (sec.) | Adam | 62 | 81 | 2721 | 1245 | 5104 |
| | | ours | 60 | 81 | 2681 | 1203 | 5049 |
| | mem. (GB) | Adam | 6.2 | 14.9 | 7.7 | 7.8 | 7.7 |
| | | ours | 6.4 | 15.8 | 8.0 | 8.0 | 8.1 |
| QWen2-0.5B | acc. (%) | Adam | 80.93 | 75.45 | 89.1 | 81.55 | 78.1 |
| | | ours | 82.84 | 77.90 | 89.81 | 83.25 | 80.43 |
| | time (sec.) | Adam | 290 | 560 | 2727 | 4652 | 15412 |
| | | ours | 267 | 503 | 2701 | 4439 | 14972 |
| | mem. (GB) | Adam | 17.7 | 35.2 | 18.7 | 10.5 | 10.5 |
| | | ours | 18.4 | 36.4 | 19.3 | 11.3 | 10.9 |
| LLaMA-3.2-1B | acc. (%) | Adam | 83.71 | 82.69 | 89.79 | 90.24 | 86.32 |
| | | ours | 85.28 | 83.37 | 90.74 | 90.88 | 88.13 |
| | time (sec.) | Adam | 335 | 1222 | 18677 | 4342 | 14743 |
| | | ours | 321 | 1193 | 18593 | 4278 | 14631 |
| | mem. (GB) | Adam | 22.4 | 45.8 | 25.5 | 25.5 | 25.5 |
| | | ours | 23.7 | 47.8 | 26.3 | 27.0 | 26.6 |

**Friedman and Nemenyi Tests.** To compare the performance of different methods, we conducted statistical tests on the accuracy results of 15 experiments involving 9 methods listed in Table 2 and Table 6. In the Friedman Test, we assume that all methods have the same average rank. Let $R_j$ denote the average rank of the $j$-th method, $k = 9$ as the number of methods, and $N = 15$ as the number of datasets. The chi-square can then be calculated as:

$$\mathcal{X}_F^2 = \frac{12N}{k(k+1)}(\sum_{j=1}^{k} R_j^2 - \frac{k(k+1)^2}{4}) \approx 63.4,$$

and the FF value is,

$$F_F = \frac{(N-1)\mathcal{X}_F^2}{N(k-1) - \mathcal{X}_F^2} \approx 15.7.$$

At the significance level of $\alpha = 0.05$, the critical value $F_\alpha = F_{0.05}(8, 14) \approx 2.699$. Since $F_F > F_\alpha$, we reject the hypothesis, indicating that there are significant differences among the methods. Next, we further perform the Nemenyi test and compute the critical distance as:

$$CD = q_\alpha \sqrt{\frac{k(k+1)}{6N}} = 3.102,$$

where $q_\alpha = 3.102$ when $k = 9$ and $\alpha = 0.05$. The visualization of the Nemenyi test, as shown in Figure 14, indicates that DOIT performs sig-

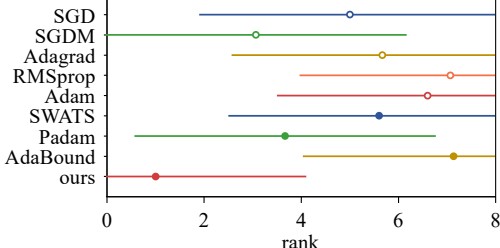

Figure 14: Visualized Nemenyi test result.

nificantly better than all other methods except SGDM and Padam. Moreover, although the differences between DOIT and SGDM/Padam are slightly below the critical distance, DOIT consistently outperforms both methods across multiple datasets. Therefore, we believe that DOIT still holds an overall performance advantage.

## F   MORE DETAILS AND RESULTS ABOUT INDEPENDENT EXPERIMENTS

Due to space limitations, we only provided a summary of the independent experiments in Section 5.3. Here, we present additional details and experimental results.

## F.1 VARYING SEARCH SPACE

To investigate the performance of DOIT under a varying optimizer search space, we conducted experiments along two dimensions: the number of optimizer types and the number of hyperparameter types. (i) In the optimizer-type experiment, we gradually expanded the search space starting from SGDM by adding SGD, Adagrad, RMSprop, ASGD, Adam, AdamW, NAdam, and Adamax. For each optimizer, the learning rate was selected from [0.1,0.01,0.001], while other hyperparameters were kept at their default values. (ii) As for the hyperparameter experiment, we systematically expanded the original HP space described in Section 5.1 (where learning rate$\in$ [0.1, 0.01, 0.001]) by including the following components: (1) weight decay for SGD, (2) momentum for SGDM, (3) weight decay for Adagrad, (4) weight decay for RMSprop, (5) alpha for RMSprop, and (6) weight decay for Adam. The ranges for these additional hyperparameters are as follows: weight decay values $\in$ [1e-2, 1e-3, 1e-4], momentum $\in$ [0.5, 0.6, 0.7, 0.8, 0.9], and alpha $\in$ [0.5, 0.6, 0.7, 0.8, 0.9].

Figure 12 illustrates the performance of DOIT compared to the baseline methods as the search space diversifies. For the baseline methods, we report the best accuracy achieved within the search space. Overall, DOIT demonstrates robust performance and consistently surpasses the baseline methods, even as the number of candidate hyperparameters increases. Furthermore, the expanded search space does not noticeably increase DOIT's time cost. As discussed in Section 4.2 and the Appendix B, the change in search space only impacts the surrogate model's prediction time within the acquisition function calculation. This overhead is minimal relative to the total training time—amounting to just $0.15\%$ even when evaluating $10^3$ configurations per search.

In addition, we conducted further experiments using Optuna-based hyperparameter optimization on ResNet18, tuning both optimizer-specific parameters (i.e.,, learning rate$\in$ $[1e-5, 1e-1]$, momentum$\in$ $[0.5, 0.99]$, weight decay$\in$ $[1e-6, 1e-2]$, and $\beta_1 \in [0.8, 0.999]$, $\beta_2 \in [0.9, 0.999]$ for Adam) and scheduler-related settings (i.e., scheduler type$\in$ {CosinAnnel, StepLR}, warm-up steps$\in$ $[0, \frac{1}{10}n_{epoch}]$, number of epochs$\in$ $[50, 100]$). As for DOIT, we search init step $n_{ini} \in$ $[10, 100]$ and switch step size $\tau \in [1e-6, 1e-2]$. As shown in Table 9, even after extensive hyperparameter tuning, DOIT consistently outperforms well-optimized single optimizers across datasets, demonstrating its robustness and the advantage of dynamic optimizer switching.

Table 9: Performance comparison under Optuna-based hyperparameter optimization on ResNet18. Results show test accuracy (%) on *usps* and *stl10*. The column "default" denotes accuracy without BO tuning, while the remaining columns report results with varying search depths (number of trials). DOIT consistently outperforms the best-tuned single optimizers.

| Dataset | Method | default | 5 | 10 | 15 | 20 | 25 |
|---------|--------|---------|-------|-------|-------|-------|-------|
| *usps*  | SGDM   | 96.10   | 95.22 | 95.86 | 95.86 | 95.86 | 96.21 |
|         | Adam   | 95.13   | 95.86 | 96.16 | 96.30 | 96.51 | 96.51 |
|         | DOIT   | 96.81   | 96.16 | 96.91 | 96.91 | 97.12 | 97.32 |
| *stl10* | SGDM   | 86.99   | 87.18 | 87.83 | 87.88 | 87.88 | 88.13 |
|         | Adam   | 76.49   | 86.81 | 86.73 | 87.87 | 87.88 | 88.03 |
|         | DOIT   | 88.23   | 88.18 | 88.19 | 88.49 | 88.83 | 88.83 |

## F.2 ABLATION EXPERIMENTS

We conducted a series of ablation studies to investigate the effectiveness of key design components.

**The selection of optimizer selection strategy and design of acquisition function.** Compared to switching optimizers with simple strategies, DOIT employs the transferability assessment $\omega_t$ and variance halving in its acquisition function. To estimate the effectiveness of our design, we compared DOIT with two simple strategies (random switch "random" and periodic replacement "cyclical") and two ablation versions (without transferability assessment "w/o $\omega_t$" and without variance halving "w/o halve"). The experimental result in Table 10 shows that simple random or periodic switching fails to produce high test accuracy. In addition, the usage of transferability assessment and variance halving both effectively enhances the adaptability of the current task, resulting in im-

proved accuracy (up to 2%) and lower variance. Additionally, we also compared DOIT with these strategies under other models (i.e., ResNet152 "RN152" and MobileNet V2 "MN"), as shown in Table 11 and Table 12.

Table 10: Experimental results for different switch strategies under full training on ResNet18.

| method | usps | | | mnist | | |
|---|---|---|---|---|---|---|
| | trainAcc. (%) | testAcc. (%) | time (sec.) | trainAcc. (%) | testAcc. (%) | time (sec.) |
| random | $98.36_{\pm 2.22}$ | $94.27_{\pm 1.41}$ | 225 | $98.91_{\pm 0.39}$ | $98.33_{\pm 0.46}$ | 1916 |
| cyclical | $97.38_{\pm 1.63}$ | $92.88_{\pm 1.94}$ | 223 | $99.49_{\pm 0.20}$ | $98.94_{\pm 0.25}$ | 1914 |
| w/o $\omega_t$ | $99.55_{\pm 0.27}$ | $95.82_{\pm 0.42}$ | 205 | $99.72_{\pm 0.09}$ | $99.25_{\pm 0.07}$ | 2101 |
| w/o halve | $99.54_{\pm 0.22}$ | $95.70_{\pm 0.35}$ | 226 | $99.67_{\pm 0.21}$ | $99.19_{\pm 0.08}$ | 2095 |
| ours | $\mathbf{99.67_{\pm 0.19}}$ | $\mathbf{96.51_{\pm 0.08}}$ | 227 | $\mathbf{99.85_{\pm 0.17}}$ | $\mathbf{99.48_{\pm 0.03}}$ | 1965 |

Table 11: The independent experimental result for transferability assessment.

| | metric | w/o $\omega_t$ | with $\omega_t$ |
|---|---|---|---|
| RN152 | trainAcc. (%) | 80.47 | **81.10** |
| | testAcc. (%) | 72.6 | **75.29** |
| | time (sec.) | 3531 | 3347+9 |
| MN | trainAcc. (%) | 98.38 | **98.42** |
| | testAcc. (%) | 91.57 | **91.93** |
| | time (sec.) | 26084 | 23271+8 |

Table 12: The independent experimental result for different optimizer switching strategies.

| | metric | random | cyclical | ours |
|---|---|---|---|---|
| RN152 | trainAcc. (%) | 76.56 | 77.79 | **81.10** |
| | testAcc. (%) | 71.49 | 71.95 | **75.29** |
| | time (sec.) | 3271 | 3299 | 3347+9 |
| MN | trainAcc. (%) | 97.58 | 98.01 | **98.42** |
| | testAcc. (%) | 90.33 | 90.73 | **91.93** |
| | time (sec.) | 24997 | 23712 | 23271+8 |

**The initial selection method.** We compared the impact of using random selection and weighted selection during the initial phase on subsequent training. In the experiments, we set the initial phase to 20 epochs and used a switch step size of $\tau = 25$. Compared with random selection, the weighted selection in DOIT significantly enhances the stability of the surrogate model, which reduces variability in the training outcomes, as shown in Figure 15.

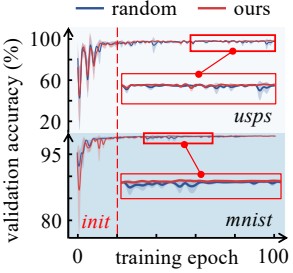

Figure 15: Ablation results of initial methods.

**The model compression technique.** Table 13 illustrates the effects of different feature compression techniques on training results. For the selected tasks (i.e., *usps* and *mnist*), simple methods such as random projection (RP) and PCA outperform the more complex UMAP, suggesting that basic compression techniques are sufficient for surrogate modeling. This is consistent with our design rationale: PCA provides stable, continuous low-dimensional projections that align well with surrogate models like GPs, while methods such as UMAP or t-SNE tend to incur higher computational costs and produce embeddings less suitable for iterative training. Together, these findings indicate that lightweight compression techniques not only reduce computational overhead but also deliver reliable performance, making them practical choices in the DOIT framework.

Table 13: Experimental results for different compression techniques under full training on ResNet18.

| method | usps | | | mnist | | |
|---|---|---|---|---|---|---|
| | trainAcc. (%) | testAcc. (%) | time (sec.) | trainAcc. (%) | testAcc. (%) | time (sec.) |
| RP | $\mathbf{99.72_{\pm 0.12}}$ | $\mathbf{96.44_{\pm 0.17}}$ | 201 | $99.61_{\pm 0.15}$ | $99.22_{\pm 0.18}$ | 1869 |
| UMAP | $98.56_{\pm 1.45}$ | $95.15_{\pm 2.15}$ | 253 | $99.73_{\pm 0.06}$ | $98.98_{\pm 0.13}$ | 2182 |
| PCA | $99.58_{\pm 0.13}$ | $96.34_{\pm 0.25}$ | 222 | $\mathbf{99.86_{\pm 0.13}}$ | $\mathbf{99.48_{\pm 0.06}}$ | 1974 |

### F.3 HYPERPARAMETER ANALYSIS

We performed an experimental analysis on the settings of hyperparameters in DOIT, including the initial step $n_{ini}$, switch step size $\tau$, and the number of PCA components. In the experiments, we set the default hyperparameter configuration to $n_{ini} = 20$, $\tau = 25$, and n_components= 2. Based on the results presented in Figure 16, we have made the following observations.

- init step $n_{ini}$: Thanks to the ongoing updates of the surrogate model during training in DOIT, even a small initial step (i.e., $n_{ini} = 20$) can produce models with high test accuracy.

- switch step size $\tau$: A smaller step size facilitates quicker switching of the optimizer, which enhances accuracy (e.g., $\tau = 20\%$ of an epoch). Conversely, a larger step size makes it more challenging to collect training data for the surrogate model, resulting in a longer switching cycle and greater variance in the results.

- PCA components number: Selecting a small number of PCA components (for example, 2) can often yield good performance in DOIT. On the other hand, using a larger number of components may impair the surrogate model's ability to learn effectively, resulting in greater variance in test accuracy.

- sub-selected layers for compression: We used only the last layer as input to the surrogate model in previous experiments. To further analyse the influence of selected layers, we constructed an independent experiment for the selected layer, as shown in Table 14. As shown in the table, adding more layers yields only marginal accuracy gains but incurs substantially higher time costs, and using too many layers may even degrade performance due to increased difficulty in surrogate modeling.

- Variance weight $\alpha$: We analyzed the sensitivity of the acquisition function to the variance weight, as shown in Table 15. On *mnist*, performance is stable across all tested values. On *usps*, accuracy is more sensitive but remains stable with moderate $\lambda$ (0.3–0.5).

- Halving period $n$: We also varied the halving period that controls how frequently the variance weight is reduced. Results in Table 16 show that larger halving periods (e.g., $1/2$ or $1/4$ of training) yield more stable performance, especially on *usps*, while *mnist* remains robust across settings.

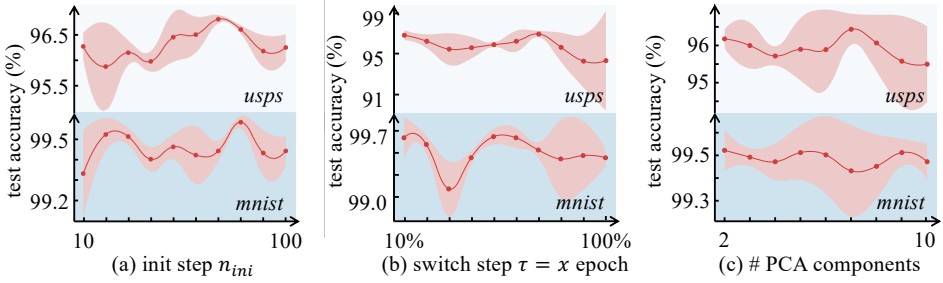

Figure 16: The experimental results of hyperparameter analysis.

Table 14: Sensitivity to the sub-selected layers for compression.

|  | only cls. | +1 layer | +2 layers | +3 layers |
|---|---|---|---|---|
| test acc. (%) | 95.02 | 95.12 | 95.07 | 95.07 |
| time (sec.) | 243 | 246 | 727 | 820 |

## G DISCUSSION ON STATE INHERITANCE

In our framework, DOIT resets optimizer states upon switching rather than inheriting them. This design choice is deliberate and generally does not cause noticeable instability. Below we provide both theoretical and empirical justification for this decision.

Table 15: Sensitivity to the variance weight $\alpha$ in the acquisition function (higher is better).

|       | 0.1   | 0.3   | 0.5   | 0.7   | 0.9   |
|-------|-------|-------|-------|-------|-------|
| *usps*  | 95.02 | 95.57 | 95.72 | 92.33 | 96.21 |
| *mnist* | 99.06 | 98.81 | 98.78 | 98.81 | 99.00 |

Table 16: Sensitivity to the halving period $n$ (fraction of training after which the variance weight is halved).

|       | /     | $\frac{1}{2}n_{epoch}$ | $\frac{1}{3}n_{epoch}$ | $\frac{1}{4}n_{epoch}$ | $\frac{1}{5}n_{epoch}$ |
|-------|-------|-----------------------|-----------------------|-----------------------|-----------------------|
| *usps*  | 89.64 | 95.22 | 94.92 | 93.42 | 95.08 |
| *mnist* | 99.22 | 99.20 | 99.21 | 99.20 | 98.62 |

- **Alignment with the core hypothesis.** DOIT selects the optimizer best suited to the current parameter state. Carrying over internal states such as momentum or running averages (e.g., transferring from Adam to SGDM) may misalign with the dynamics of the new optimizer, undermining the purpose of switching.

- **Empirical evidence.** We compared DOIT with and without state inheritance across multiple datasets (e.g., transferring momentum between Adam and SGDM). As shown in Table 17, simple inheritance strategies do not consistently improve performance and may even lead to degradation or instability.

- **Technical complexity.** Robust state transfer across optimizers with different internal dynamics (e.g., momentum in SGD vs. second-order statistics in Adam) remains an open problem. Designing such mechanisms goes beyond the scope of this paper, but we consider it an important direction for future work.

- **Training dynamics of DOIT.** As analyzed in Sec. 5.2 and visualized in Figs. 9, 11, and 13, any instability due to state resetting primarily occurs in the early phase of training, when the surrogate model is still adapting. Later, optimizer selection stabilizes and no significant loss spikes are observed. In fact, resetting states can sometimes mitigate instability by preventing overly aggressive updates (e.g., momentum inheritance leading to divergence).

Table 17: Comparison of DOIT with and without state inheritance.

| Method        | usps  | mnist | stl10 |
|---------------|-------|-------|-------|
| State inherit | 95.02 | 99.38 | 77.13 |
| State reset   | 96.11 | 99.15 | 88.11 |

## H LLM USAGE STATEMENT

Large language models (LLMs) were only used in this work to assist with writing, including language polishing and grammar checking.

