# OpenReview forum: "Towards Dynamic Interleaving Optimizers"
_ICLR.cc/2026/Conference — ICLR 2026 Poster_

### Official Review · Reviewer_ffH8 · 2025-10-25

**Soundness:** 3
**Presentation:** 2
**Contribution:** 3
**Rating:** 6
**Confidence:** 4

**Summary:**

This paper introduces DOIT (Dynamic Optimizer Interleaving Training), a novel method that dynamically switches optimizers during training. The core hypothesis is that different optimizers are better suited for different training states. DOIT models this problem by building Gaussian Process (GP) surrogate models to predict the "short-term" performance of different optimizers based on a PCA-compressed representation of the current parameter state. A well-designed acquisition function—which considers predicted performance, uncertainty, task transferability, and training progression—is used to select the optimal optimizer for the next training cycle.

The authors provide extensive empirical validation across a wide range of tasks (image classification, text classification, machine translation, object detection), models (ResNet, ViT, RoBerta), and settings (full training, PEFT). The results consistently show that DOIT achieves higher accuracy (1-3% gains) and faster convergence (2-10% faster) compared to a large suite of static and simple-hybrid optimizers.

**Strengths:**

1.The motivation of this paper is direct and clear. The paper presents a principled approach to optimizer selection. Moving from static selection or simple two-stage hybrids (like SWATS) to dynamic, state-aware selection framework is a significant conceptual leap.

2.The breadth of tasks, datasets, and models is comprehensive and convincing. The inclusion of ablation studies, cost analysis, and case studies on switching behavior thoroughly validates the method's design and effectiveness.

3.Initially i questioned if the overhead of training multiple GP online would be significant. However, the authors provide a detailed cost analysis and empirical measurements suggesting that the overhead from surrogate modeling and selection is minimal (often <1%), making the method practical.

**Weaknesses:**

1.The most significant weakness is that the method resets optimizer states (e.g., momentum, second-moment estimates) upon every switch. This seems counter-intuitive and potentially disruptive, as the optimizer loses all accumulated history. The authors defend this in Appendix, arguing that state inheritance is complex and their empirical tests (Table 15) showed it can destabilize training. However, this should be further proven.

2.The surrogate model's input is a PCA projection of some model layers. The paper is not perfectly clear on which layers are chosen for the different models, and how sensitive the performance is to this choice (e.g., classifier only vs. final N blocks).

**Questions:**

1.Some pervious work also explore the potential of dynamic hybridization of different optimizer, such as HUB (hybrid-update-based optimization strategy). They emphasized the hybridization with learned optimizers. Could you testify if DOIT could also handle the switching timing for those optimizers?

2. How was the subset of layers for PCA compression chosen for the main experiments (e.g., for ViT and RoBerta)? How sensitive is DOIT's performance to this choice?

3.The acquisition function includes a transferability weight $\omega_t$. Does this imply the surrogate models must be re-trained from scratch for every new task, or can the GP models themselves be transferred?

---

> ### Author Response · Authors · 2025-11-19
> **Response**
>
> Thank you for your thoughtful and detailed review. We have thoroughly revised our draft based on your constructive comments. Below, we provide our response to your suggestion point-by-point.
>
> ---
>
> > **W1. DOIT resets the optimizer state.**
>
> We understand the reviewer’s concern regarding resetting optimizer states, and we also acknowledge that state inheritance may have potential value (cf. Limitations and future work). As discussed in Appendix G, this issue involves both technical difficulties and empirical considerations. Here we provide an intuitive explanation.
>
> For example, as shown in Fig. 11 (the case of switching between SGD and SGD with momentum), inheriting momentum causes the update direction still guided by its historical momentum, preventing the switch from producing the benefits. This contradicts DOIT’s core goal of selecting the optimizer best suited to the current parameter state. In contrast, resetting the state ensures that each switch reflects the current training dynamics, allowing the newly selected optimizer to operate without being biased by incompatible historical statistics.
>
> Resetting states may preserves the effectiveness of dynamic switching, while robust cross-optimizer state transfer is a more complex open problem that we leave for future work.
>
> ---
>
> > **W2 & Q2. The selected layer in PCA compression.**
>
> Thank you for your reminder. In all experiments, we used only the last layer as the input to the surrogate model, and we have clarified it in the revised submission. To further justify this choice, we added sensitivity studies in Appendix F.3 (Table 14) by varying the number of layers included in the PCA-based representation. The results show that incorporating additional layers (e.g., the final few blocks) leads to only marginal accuracy improvements while substantially increasing the computational cost of PCA and surrogate updates. Moreover, using too many layers can even degrade performance due to the increased difficulty of surrogate modeling in a higher-dimensional input space. Therefore, using the last layer provides a favorable balance between performance and computational efficiency.
>
> | **Layers  used** | **only  cls.** | **+1  layer** | **+2  layers** | **+3  layers** |
> | ---------------- | -------------- | ------------- | -------------- | -------------- |
> | test  acc. (\%)  | 95.02          | 95.12         | 95.07          | 95.07          |
> | time  (sec.)     | 243            | 246           | 727            | 820            |
>
> ---
>
> > **Q1. Could DOIT also handle the switching timing for the methods which explore the hybridization of different optimizer (e.g., HUB).**
>
> Theoretically, DOIT could incorporate learned optimizers as candidates. However, this is not ideal in practice because DOIT has a different motivation with hybrid methods. The hybrid methods (e.g., HUB[1] and LGL2O[2]) combine hand-designed gradient rules with learned optimizers to address the inherent instability and poor out-of-distribution robustness of purely learned optimizers. In contrast, DOIT aims to leverage the strengths of different optimizers across training stages rather than fixing the weaknesses of any single optimizer. Extending DOIT to learned optimizers would require additional investigation into their stability and context dependence. We have added related explanations and citations in the revised submission.
>
> [1] Dai G, Wu W, Wang Z, et al. HUB: Enhancing Learned Optimizers via Hybrid Update-based Strategy[J].
>
> [2] Prémont-Schwarz I, Vítků J, Feyereisl J. A simple guard for learned optimizers[J]. arXiv preprint arXiv:2201.12426, 2022.
>
> ---
>
> > **Q3. The acquisition function includes a transferability weight $\omega_t$. Does this imply the surrogate models must be re-trained from scratch for every new task, or can the GP models themselves be transferred?**
>
> The transferability weight $\omega_t$ models how well a pretrained model initially adapts to the current task and is unrelated to cross-task transfer. Our GP surrogate is updated online within each task and is not reused across tasks, as parameter and gradient distributions differ substantially between tasks. Given the low computational cost of training the GP surrogate (Appendix B), rebuilding it per task is both lightweight and appropriate.
>
> ---

---

> > ### Comment · Reviewer_ffH8 · 2025-11-19
> > **Reply for Author**
> >
> > I have carefully reviewed the authors' responses and acknowledge that certain questions may pertain to broad topics that are challenging to fully address within a single manuscript. Nevertheless, I appreciate the effort and rigor the authors have demonstrated in this work.
> >
> > As previously mentioned, DOIT is a well-motivated method with a clear focus. Each design component appears to contribute meaningfully to the overall performance, without unnecessary complexity—this clarity is valuable. While I must note that my familiarity with recent advances in this specific area is limited, I maintain my original evaluation score but am inclined to assign a higher level of confidence. I hope the authors find this assessment reasonable.
> >
> > Reviewer ffH8

---

> > > ### Author Response · Authors · 2025-11-20
> > > **Thank you for your thoughtful and constructive review.**
> > >
> > > Thank you for your positive evaluation and appreciation of our responses. We find your feedback encouraging, and we greatly appreciate the insights you have provided. Exploratory directions, such as optimizer state inheritance and hybrid strategies with learned optimizers, are our upcoming priorities for this research series. Thank you again for the time and effort you have dedicated to our work.
> > >
> > > Authors of submission #3551

---

### Official Review · Reviewer_MSZS · 2025-10-28

**Soundness:** 2
**Presentation:** 3
**Contribution:** 2
**Rating:** 4
**Confidence:** 4

**Summary:**

This paper proposed a new framework called Dynamic Optimizer Interleaving Training (DOIT) for training deep neural networks. The main idea is to use a surrogate model, more specifically, a Gaussian Process (GP), to assess the performance of the optimizers, and then dynamically select the best one from the candidates. Experiments on several datasets have been conducted to demonstrate the effectiveness of the proposed method.

**Strengths:**

1. The paper is well-written and easy to follow.
2. The performance of the proposed method seems good.

**Weaknesses:**

1. The novelty of this work is not significant enough. Specifically, the key component of the proposed framework is a surrogate model constructed by GP, which, however, is not unusual and has been widely used in hyperparameter optimization. Despite this key component, other designs are mostly engineering and heuristic, such as the performance score and the acquisition function.

2. There is no theoretical convergence guarantee for the proposed method. Since the training process by the proposed framework includes the switch of multiple optimizers and corresponding hyperparameters, the convergence behavior could be more complex than a single optimizer, and thus, a theoretical convergence guarantee can be useful to guide practice. Besides, it would also be interesting to analyze how good the surrogate function is and how it could affect the final performance.

**Questions:**

First, the authors should address the issues mentioned in the "Weaknesses". Despite these weaknesses, several more things should be clarified:

1. The authors only report the FLOPs of the proposed method, but it is also important to consider the memory overhead of the additional operations, such as PCA compression and GP prediction.

2. It seems that the optimizer configuration candidates lie in a discrete space, and thus, it could be tricky to construct the candidates.

---

> ### Author Response · Authors · 2025-11-19
> **Response**
>
> Thanks for your thoughtful and constructive suggestions. We have thoroughly revised our submission based on your comments. Below is our response to your suggestion.
>
> ---
>
> > **W1. Novelty concerns. The GP surrogate is widely used in hyperparameter optimization.**
>
> Although DOIT adopts the idea of surrogate models and acquisition functions from SMBO (e.g., Bayesian Optimization), it differs fundamentally in problem definition and mechanism, which is also the main novelty of our work. Traditional GP surrogate is used for static hyperparameter optimization, where the surrogate models the static mapping from a fixed configuration to final performance. In contrast, DOIT focuses on the dynamic training process, learning a mapping from the current parameter landscape to the short-term benefits of different optimizers and making decisions online throughout training. Moreover, the design of DOIT’s input features and acquisition objective is tailored to training dynamics instead of static HPO. We clarify these relationships and distinctions in Sec. 4.1.
>
> ---
>
> > **W2. Concerns about convergence guarantee under optimizer switching and the influence of surrogate model quality.**
>
> Although DOIT performs optimizer switching during training, its overall convergence behavior is guaranteed by the underlying optimizers. DOIT does not modify any optimizer’s update rule and instead selects from standard optimizers at different stages, so the convergence behavior aligns with that of the chosen optimizer at each stage. Moreover, the acquisition function encourages DOIT to favor more stable optimizers in the later phases of training (as shown in Sec. 5.2), ensuring a stable and reliable convergence. We have added further discussion in Sec. 4.2 and Appendix B. Regarding the impact of the surrogate model, we analyze and discuss it through sensitivity studies on input-related hyperparameters (such as the compression method and the number of PCA components) in Appendix F.3.
>
> ---
>
> > **Q1. Memory overhead of additional operations.**
>
> We provide a theoretical analysis of memory overhead in Appendix C, where the additional cost mainly comes from storing historical records and surrogate models, accounting for only about 2.5\% of the total training memory on ResNet18. In addition, we further include empirical measurements of memory usage in Table 8 (Appendix E), which are consistent with the theoretical analysis. Across models such as GPT-2, DOIT introduces less than 6% additional memory overhead.
>
> | model | metric    | method | mrpc | rte  | qqp  | qnli | mnli |
> | ----- | --------- | ------ | ---- | ---- | ---- | ---- | ---- |
> | GPT2  | mem. (GB) | Adam   | 6.2  | 14.9 | 7.7  | 7.8  | 7.7  |
> |       |           | ours   | 6.4  | 15.8 | 8.0  | 8.0  | 8.1  |
> | Qwen  | mem. (GB) | Adam   | 17.7 | 35.2 | 18.7 | 10.5 | 10.5 |
> |       |           | ours   | 18.4 | 36.4 | 19.3 | 11.3 | 10.9 |
> | LLaMA | mem. (GB) | Adam   | 22.4 | 45.8 | 25.5 | 25.5 | 25.5 |
> |       |           | ours   | 23.7 | 47.8 | 26.3 | 27.0 | 26.6 |
>
> ---
>
> > **Q2. Candidate construction is tricky due to the discrete search space in DOIT.**
>
> In DOIT, the optimizer candidates are not restricted to a discrete set. While the optimizer types themselves form a discrete space, the hyperparameters within each optimizer (e.g., momentum, learning rate, weight decay) lie in continuous domains and are directly used as inputs to the surrogate models. As a result, DOIT effectively operates in a mixed space consisting of discrete optimizer types and continuous hyperparameters.
>
> The candidate configurations are generated through continuous sampling rather than predefined discretization, with the number of trials controlling the exploration. Appendix F.1 (cf. Table 9) provides experimental results demonstrating that DOIT can effectively construct and select candidates even in complex configuration spaces.
>
> ---

---

### Official Review · Reviewer_5cJs · 2025-10-31

**Soundness:** 3
**Presentation:** 3
**Contribution:** 3
**Rating:** 6
**Confidence:** 3

**Summary:**

This paper introduces a novel method called Dynamic Optimizer Interleaving Training (DOIT), which dynamically switches between optimizers during training to enhance model performance. Traditional training processes typically rely on a single static optimizer or a simple hybrid, missing opportunities to adapt to changing dynamics during training. DOIT builds surrogate models to predict optimizer performance based on the current parameter states and selects the most suitable optimizer using an acquisition function. Experiments across tasks such as classification, translation, and object detection show that DOIT improves convergence speed (2% to 10%) and accuracy (1% to 3%). Further case studies validate its effectiveness.

**Strengths:**

1. The paper proposes a dynamic optimizer to improve training performance.

2. The paper provides comprehensive experiments in both the vision and NLP domains, validating the proposed method's effectiveness.

3. The paper includes a comparison of training efficiency, which is quite useful.

**Weaknesses:**

1. The method requires retaining multiple optimizers, which may lead to high memory usage. The authors should include a direct comparison to address this concern.

2. As mentioned by the authors, the method does not support optimizers with momentum. Since momentum is a key mechanism in current mainstream optimizers, I recommend that the authors propose a feasible solution to extend their method to support momentum-based optimizers.

3. While some details still need clarification, I understand that the method retains multiple optimizers and uses strategies like performance-aware weighting after a certain number of steps (Eq. 6) for weight updates. In this process, where exactly is the "training dynamic" reflected? Additionally, doesn’t this approach resemble ensemble learning? Furthermore, the method seems related to meta-learning; it would be helpful for the authors to analyze and compare these connections.

**Questions:**

N/A

---

> ### Author Response · Authors · 2025-11-19
> **Response**
>
> Thank you for your thoughtful and constructive review. We have carefully revised the manuscript and addressed each of your concerns in detail. Below, we provide point-by-point responses to your comments.
>
> ---
>
> > **W1. Direct comparison of memory usage.**
>
> We provide a theoretical analysis of memory overhead in Appendix C, where the additional cost mainly comes from storing historical records and surrogate models, accounting for only about 2.5\% of the total training memory on ResNet18. Furthermore, we added results in the revised draft (cf. Table 8, Appendix E) to present empirical memory usage. For example, on GPT-2, DOIT introduces only about 5\% additional memory overhead.
>
> ---
>
> > **W2. Support for optimizers with momentum.**
>
> Our method does not prohibit the use of momentum-based optimizers. The only restriction is that when switching between optimizer types (e.g., from SGD with momentum to Adam), DOIT does not inherit the momentum state from the previous optimizer (cf. Sec. 6 and Appendix G). This design choice is due to the incompatibility of internal states across different optimizers, and inheriting such states may lead to unstable updates. We provide additional discussion regarding the feasibility of transferring optimizer states in Appendix G.
>
> ---
>
> > **W3. Clarify “training dynamic”, and distinguish DOIT from ensemble learning and meta-learning.**
>
> The “training dynamic” refers to the evolving parameter landscape during training, where different optimizers become more or less effective at different stages. DOIT captures this behavior by modeling each optimizer with a surrogate and selecting the most appropriate one online through the acquisition function.
>
> DOIT and ensemble learning are fundamentally different. Ensemble methods (e.g., GBDT, Random Forest) focus on combining predictions from multiple models, whereas DOIT does not train or combine multiple models. Instead, it performs dynamic optimizer selection for a single model across different training phases, which is a temporal decision process rather than a model ensemble.
>
> As for meta-learning, DOIT is related in that it performs a form of meta-level optimizer selection, similar to AutoML. However, meta-learning typically focuses on cross-task adaptation, whereas DOIT performs lightweight, online optimizer switching within a single training run. We discuss these relationships in Sec. 2 and have further clarified the distinctions in the updated version.
>
> ---

---

### Official Review · Reviewer_PbZG · 2025-11-01

**Soundness:** 2
**Presentation:** 3
**Contribution:** 2
**Rating:** 4
**Confidence:** 4

**Summary:**

This paper introduces dynamic interleaving of optimizers (DOIT) framework that adaptively switches among multiple optimizers during training based on the training state. The method constructs and continually updates Gaussian surrogate models for each optimizer to estimate their expected performance under different conditions. At each switching step, the optimizer predicted to yield the highest improvement is selected for the next training phase.

**Strengths:**

- Novel approach for interleaving optimizers during training based on surrogate quantitative performance
- Experiments are conducted on a variety of datasets (6 CV, 2 NLP and 3 additional tasks) with two settings: full training and fine-tuning setups and compared against standard optimizer baselines (although a bunch of these tasks are small as per today's standards, see weaknesses below)

**Weaknesses:**

- The approach introduces a lot of additional hyper-parameters (see questions below). Moreover, optimal DOIT hyperparameters are different per dataset which limits the pratical utility of the approach and introduces additional tuning overhead for the practitioner.
- The proposed approach incurs high memory and runtime overhead which are reduced by arbitrary heuristics such as PCA over parameters, additional sub-selection of layers (only last). It's unclear if these heuristics will hold for large-scale ML workloads. Similarly, acquisition function is hand-designed with heuristics. These are central components of the proposed approach and non-trivial changes to the approach may be required to make it scalable to standard ML tasks on which optimizers are usually benchmarked (GPT-2, ViT, ResNet50, AlgoPerf benchmark, etc).
- Experiment datasets are small as per today's standard (4/6 CV datasets are mnist-scale, imagenet is the only relevant one) and also heavily biased towards vision tasks (6 CV datasets, coco object detection)
- The approach requires extra surrogate model (Gaussian processes) which greatly limits scalability of the proposed approach (n^3 complexity).
- In order to collect experience data for surrogate model, training is performed with random optimizer configurations -- wouldn't this lead to irreversible damage to the parameters of the model which may be hard to recover from at later stages in the training?
- While DOIT switching is mainly guided by the short-term benefits of each optimizer, practitioners aiming for highly efficient and performant training also consider long-term effects, taking into account the total number of iterations, final accuracy, and selecting hyperparameters such as learning rate, weight decay, and their schedules accordingly.

**Questions:**

Some questions on hyperparameters and heuristics:
- how long is the experience collection phase per dataset and what % of training is dedicated to it? is it a tunable parameter?
- what are hyperparameters of gaussian processes model? how are they chosen for each dataset/task?
- how is switch cycle tau chosen? Is it an extra tuning parameter?
- which layers are sub-selected for surrogate model for each dataset?
- for a given selected optimizer in a switch cycle based on acquisition function score, how are optimizer hyper-parameters chosen (such as learning rate, momentum, weight decay, betas if applicate etc)?

Moreover, can authors please elaborate on the key runtime/memory bottlenecks in the current approach? What are the most and least expensive parts in the approach and how do they change (if they do) based on DOIT hyperparameters?

---

> ### Author Response · Authors · 2025-11-19
> **Response [1/2]**
>
> Thank you for your thoughtful and constructive review. We have thoroughly revised our draft based on your comments. Below, we provide our response to your suggestion point-by-point.
>
> ---
>
> > **W1. Hyperparameter tuning cost.**
>
> DOIT introduces several additional hyperparameters, but their practical tuning cost is small. These hyperparameters have stable default values and clear effective ranges, so they do not require dataset-specific adjustment. Specifically: (1) **The default configuration already works consistently across datasets and models.** As shown in Tables 2–4, we use a unified default setting for all experiments without dataset-specific tuning, yet still obtain stable improvements. (2) **Sensitivity studies reveal well-behaved and controllable effective ranges.** In Sec. 5.3 and Appendix F.3, we provide additional ablation and sensitivity studies showing that each hyperparameter has a well-behaved effective range, from which practical recommended settings can be derived (e.g., $\tau=10\%\times$ an epoch). (3) For the specific hyperparameters highlighted by the reviewer, we have added the corresponding experiments and clarifications in Appendix F (see R7–R10).
>
> ---
>
> > **W2. Heuristic overhead and scalability.**
>
> Our method introduces a controllable memory and runtime overhead. As analyzed in Sec. 4.2 and Appendix B–C, the runtime overhead is below 1\% and the memory overhead is below 2.5\%. For the heuristic components (e.g., PCA projection, layer sub-selection, and the acquisition function), we provide corresponding ablation and sensitivity studies in Sec. 5.3 and Appendix F.2–F.3, which verify their effectiveness and stability.
>
> Regarding scalability, DOIT consistently improves performance across a variety of larger models, e.g., +1.2% test accuracy on ViT (ImageNet, Table 2) and over +1% on ResNet152 (Table 7). In the revised manuscript, we further include experiments on GPT-2, LLaMA, and QWen (as shown in the following table, detailed in Appendix E), which show consistent gains, demonstrating that DOIT remains robust across diverse architectures and scales.
>
> |    |     | **mrpc** |    |    | **rte**  |    |    | **qqp**  |    |    |
> | - | - | - | - | - | - | - | - | - | - | - |
> | **model** | **method** | **acc.** (\%) | **time** (sec.) | **mem.** (GB) | **acc.** | **time** | **mem.** | **acc.** | **time** | **mem.** |
> | GPT2      | Adam          | 71.43    | 62       | 6.2      | 58.35    | 81       | 14.9     | 86.02    | 2721     | 7.7      |
> | | ours      | 72.23         | 60       | 6.4      | 59.13    | 81       | 15.8     | 87.48    | 2681     | 8.0      |
> | QWen      | Adam          | 80.93    | 290      | 17.7     | 75.45    | 560      | 35.2     | 89.1     | 2727     | 18.7     |
> | | ours      | 82.84         | 267      | 18.4     | 77.90    | 503      | 36.4     | 89.81    | 2701     | 19.3     |
> | LLaMA     | Adam          | 83.71    | 335      | 22.4     | 82.69    | 1222     | 45.8     | 89.79    | 18677    | 25.5     |
> | | ours      | 85.28         | 321      | 23.7     | 83.37    | 1193     | 47.8     | 90.74    | 18593    | 26.3     |
>
> ---
>
> > **W3. Dataset size and domain diversity.**
>
> Our experiments were designed to cover datasets of different scales and task types, rather than focusing solely on small CV benchmarks. The current draft includes 6 vision classification datasets (including ImageNet and ImageNet-A), COCO detection, WMT14 translation, EUNITE regression, and two NLP tasks (MRPC and QQP). In the revised version, we additionally include results on GPT, LLaMA, and QWen models evaluated on RTE, QNLI and MNLI (Table 8, Appendix E). Overall, DOIT shows fairly consistent performance across these diverse domains and dataset sizes, suggesting a certain level of generality.
>
> ---
>
> > **W4. Scalability bottleneck from GP ($n^3$ complexity).**
>
> The GP surrogate does not limit the scalability of our approach. Its computational complexity is determined by the number of switching steps rather than the model size. Specifically, the $n^3$ term in GP corresponds to the number of “switching rounds” used for surrogate fitting, which is directly controlled by the switching ratio within each epoch (e.g., $\tau = 10\% \times$ an epoch). As a result, the surrogate cost does not increase with model parameter size or dataset scale. More detailed analyses of time and memory overhead are provided in Sec. 4.2 and Appendix B-C.
>
> ---
>
> > **W5. Risk of parameter damage from initial random optimizers.**
>
> The random initial phase does not cause irreversible effects on the model parameters. It occupies only a small portion of training (e.g., 1.6%) and uses a small learning rate, so the perturbation is minimal. As shown in Fig. 10 and Fig. 13, although random switching causes brief fluctuations, the model quickly returns to a stable training trajectory. In addition, as discussed in Sec. 5.2, the optimizer choices become increasingly stable after the initial phase, and training proceeds stable.
>
> ---

---

> > ### Author Response · Authors · 2025-11-19
> > **Response [2/2]**
> >
> > ---
> >
> > > **W6. DOIT focuses on short-term benefits, but long-term effects are also important.**
> >
> > Although DOIT’s surrogate output focuses on short-term benefits, the acquisition function accounts for variance, transferability, and training progress, effectively incorporating longer-term effects. For example, as shown in Sec. 5.2, DOIT tends to prefer faster optimizers (e.g., Adam) early on and more stable ones (e.g., SGD) later, reflecting its inherent long-term optimization behavior.
> >
> > ---
> >
> > > **Q1. How long is the experience collection phase per dataset and what % of training is dedicated to it? is it a tunable parameter?**
> >
> > In the overall experiments of Sec. 5.1, we use a unified setting of $n_{ini}=50$ switching cycles for the experience-collection phase, with each switch having $\tau=25$ iterations. Its percentage of the total training is computed as
> > $\frac{\tau\cdot n_{ini}}{n_{EPOCH} \cdot N / n_{BZ}}$, where $N$ is the number of instance and $n_{BZ}$ is the batch size. For example, this accounts for approximately 1.3\% of the training steps on MNIST and 1.6\% on CIFAR-10. This is a tunable hyperparameter, and a detailed discussion is provided in Appendix F.3.
> >
> > ---
> >
> > > **Q2. What are hyperparameters of gaussian processes model? how are they chosen for each dataset/task?**
> >
> > We implement the surrogate model using `GaussianProcessRegressor` from sklearn with its default hyperparameter settings (including the default kernel and regularization). This default configuration worked stably across all tasks, so no additional per-task tuning was applied. We have added this clarification to Appendix D.
> >
> > ---
> >
> > > **Q3. How is switch cycle tau chosen? Is it an extra tuning parameter?**
> >
> > We set the switch cycle to the default value of $\tau = 25$ iterations in our experiments, and it is a tunable hyperparameter. This default setting works well across different tasks, so no additional tuning was applied. Further discussion of this hyperparameter and the sensitivity analysis is provided in Fig. 16 (b), Appendix F.3.
> >
> > ---
> >
> > > **Q4. Which layers are sub-selected for surrogate model for each dataset?**
> >
> > We used only the last layer as input to the surrogate model in all experiments. To further examine this choice, we have added sensitivity studies in Appendix F.3 by varying the number of layers used for the PCA-based representation. As shown in the following table, adding more layers yields only marginal accuracy gains but incurs substantially higher time costs, and using too many layers may even degrade performance due to increased difficulty in surrogate modeling.
> >
> > | **Layers  used** | **only  cls.** | **+1  layer** | **+2  layers** | **+3  layers** |
> > | ---------------- | -------------- | ------------- | -------------- | -------------- |
> > | test  acc. (\%)  | 95.02          | 95.12         | 95.07          | 95.07          |
> > | time  (sec.)     | 243            | 246           | 727            | 820            |
> >
> > ---
> >
> > > **Q5. For a given selected optimizer in a switch cycle based on acquisition function score, how are optimizer hyper-parameters chosen (such as learning rate, momentum, weight decay, betas if applicate etc)?**
> >
> > DOIT constructs a surrogate model per optimizer type, and the input to each surrogate includes the corresponding hyperparameter vector $\boldsymbol{\lambda}^i$ (cf. Sec. 3.3). During training, DOIT evaluates every “optimizer + hyperparameter” candidate by computing its acquisition function score and selects the combination with the highest score, thereby determining both the optimizer type and its hyperparameters.
> >
> > ---
> >
> > > **Q6. Moreover, can authors please elaborate on the key runtime/memory bottlenecks in the current approach? What are the most and least expensive parts in the approach and how do they change (if they do) based on DOIT hyperparameters?**
> >
> > We provide a detailed analysis of DOIT’s runtime and memory overhead in Sec. 4.2 and Appendix B-C. A brief summary is as follows.
> >
> > - **Runtime.** The additional runtime in DOIT mainly comes from three components: transferability assessment, surrogate model updating, and acquisition calculation. Among them, surrogate updating is the dominant cost ($\approx 0.8\%$ on ResNet18 + CIFAR10), followed by acquisition calculation (0.15\%), while transferability assessment contributes the least (0.03\%). These costs primarily depend on the number of optimizer switches (controlled by $\tau$) and the PCA dimension. Detailed formulas and numerical examples are given in Appendix B.
> >
> > - **Memory.** The extra memory usage mainly arises from (1) storing compressed parameter vectors and performance scores, and (2) maintaining the Gaussian process kernel matrices. Both depend on the number of switching steps and the PCA dimension (Appendix C). Overall, the memory overhead is small, for example, on ResNet18 it amounts to roughly 12.5% of the parameter size (less than 3\% of vanilla training). Additional empirical evidence is provided in Appendix E Table 8 (see R2).
> >
> > ---

---

### Author Response · Authors · 2025-11-19
**Global Response**

Dear Area Chair, SAC, and Reviewers,

We appreciate your dedicated time and effort in reviewing our submission. The valuable comments help us improve our submission. We have revised the manuscript and are now submitting the revised pdf along with this summary of the revisions. To facilitate differentiation, we have highlighted key modiﬁcations in the updated pdf in blue. The main changes in the revised submission are as follows.

- **Experiments:** We have added additional evaluations on more NLP models (GPT-2, QWen, LLaMA) and tasks (RTE, QNLI, MNLI), reporting results on test accuracy, convergence speed, and peak memory. We also included sensitivity studies on the choice of layers for compression (cf. Table 14, Appendix F.3).

- **Methodology:** We have refined the discussions on convergence, efficiency, and memory overhead (cf. Sec. 4.2). In addition, we have clarified the connections and distinctions between DOIT, meta-learning, and hybrid optimizers (cf. Sec. 2).

- **Presentation:** We have corrected several typographical errors and improved the clarity and readability of several descriptions.

Sincerely,

Authors of submission #3551

---

### Author Response · Authors · 2025-12-03
**Summary of Revisions and Context for the Area Chair**

Dear Area Chair,

Thank you for stepping in during this unexpected transition. Below we provide a brief summary of our work and the revisions made during the rebuttal, which we hope will be helpful as you make the assessment of our submission.


**Summary.** Training progresses through stages with shifting optimization needs, while standard practice still relies on a single optimizer. To better accommodate this dynamic process, we propose DOIT, a dynamic optimizer selection framework that uses surrogate models and a well-designed acquisition function to select the optimizer suited for each stage. DOIT achieves faster convergence (2%-10% faster) and higher accuracy (1%-3% improvement) across diverse tasks and models.


**Main updates during rebuttal.**

1.	**Expanded experiments on more models and tasks.** We have added additional evaluations on more NLP models (GPT-2, QWen, LLaMA) and tasks (RTE, QNLI, MNLI), reporting results on test accuracy, convergence speed, and peak memory.

2.	**Detailed runtime and memory overhead analysis.** We have added detailed runtime and memory cost discussions, with empirical measurements confirming minimal overhead in practice (typically <1% runtime, <2.5% memory).

3.	**More hyperparameter studies.** We have added sensitivity studies on the choice of layers for compression.

4.	**Clarified methodological descriptions.** We have refined the discussion part and improved the clarity of several descriptions.


During the rebuttal period, Reviewer ffH8 provided a follow-up response indicating that our clarifications were helpful and **increased their confidence rating (4->5) while maintaining a positive assessment.** Thank you again for your time and effort, and we hope this context could be helpful as you evaluate the submission.


Sincerely,

Authors of submission #3551

---

### Meta-Review · Area_Chair_PKLF · 2026-01-05

**Summary:**

The submission proposes a method for dynamically switching between neural network optimisers through the use of Bayesian optimisation with Gaussian Processes. The reviewers largely appreciated the strong results, clear presentation, and acknowledge that this represents the first use of Bayesian optimisation for this problem. The reviewers raised a number of concerns related to the practicality of the proposed approach, centring mostly on the introduction of new hyperparameters and the additional overhead incurred by the method. Concerns about the scope of the experimental evaluation were also raised.

There were a few comments phrased as weaknesses by reviewers that are really just hypotheticals (e.g., potential suboptimalities or suggestions of other algorithm design choices). While these are interesting points to discuss, they are not weaknesses in the paper and I therefore did not consider them when making my decision.

**Reviewer Concerns:**

The authors have done a good job of providing additional evidence during the rebuttal phase showing that the concerns related to new hyperparameters and compute/memory overhead do not lead to any significant problems. The new experiments on more language datasets also go some way to addressing the concern related to the scope of the evaluation.

**Reviewer Scores:**

I expect that reviewers MSZS and PbZG would have changed their currently borderline negative score to a positive score.

---

### Decision · Program_Chairs · 2026-01-26

Accept (Poster)